# Development and validation of a hybrid data-driven model-based wake steering controller and its application at a utility-scale wind plant

Peter Bachant[1], Peter Ireland[1], Brian Burrows[1], Chi Qiao[1], James Duncan[1], Danian Zheng[1], and Mohit Dua[1]

[1]WindESCo, Inc., 265 Franklin St, Boston, MA 02110, USA

**Correspondence:** Peter Bachant (pete@windesco.com)

**Abstract.** Despite the promise of wind farm control through wake steering to reduce wake losses, the deployment of the technology to wind plants has historically been limited to small and simple demonstrations. In this study, we develop and deploy a wake steering control system to 10 turbines within a complex 58 turbine wind plant. A multi-month data collection campaign was used to develop a closed-loop tuning and validation process for the eventual deployment of the system to 165 turbines on this and two neighboring wind plants. The system employs a novel actuation strategy, using absolute nacelle position control instead of yaw sensor offsets, along with a model in the loop performing real-time prediction and optimization. The novel model architecture, which employs data-driven input estimation and calibration of an engineering wake model along with a neural network-based output correction, is examined in a validation framework that tests predictive capabilities in both a dynamic (i.e., time series) and aggregate sense. It is demonstrated that model accuracy can be significantly increased through this architecture, which will facilitate effective wake steering control in plant layouts and atmospheric conditions whose complexities are difficult to resolve using an engineering wake model alone.

## 1 Introduction

Wind turbines extract kinetic energy and momentum from the atmosphere to convert into electrical energy. This process generates a wake downstream of each turbine, where the wind speed is slower and more turbulent than the inflow. As wind turbines are commonly deployed in clusters or arrays, the wake of one turbine can negatively impact the production of neighboring downstream turbines. Wind plant wake losses are site-specific, depending on both the wind climatology (e.g., wind speed, direction, turbulence) and the wind plant layout (e.g., turbine row spacing). Wake losses for US wind plants, both onshore and offshore, have been estimated to be between 2 and 20% (Bensason et al., 2021; Lee and Fields, 2021), with offshore wake losses expected to be considerably higher than onshore losses (Rosencrans et al., 2023).

Modern concepts to enable wind farm flow control, where turbines in a wind plant work collectively to maximize overall wind plant production, as opposed to "greedy" individual turbine-centric control behavior, have opened up avenues for wake loss mitigation. Recovering as little as 1% in annual energy production (AEP) through wake mitigation can provide on the order of an additional billion dollars in annual revenue (assuming 1 TWh/yr production at $0.10/kWh (EIA)) for an industry that

is increasingly subject to margin pressure. Because of this immense opportunity, methods for wake mitigation have recently received significant research attention (Dong et al., 2022; Andersson et al., 2021).

One promising method for wake mitigation is wake steering, which involves strategic yaw misalignment to redirect turbine wakes away from neighboring downstream turbines (Howland et al., 2022b; Campagnolo et al., 2022; Doekemeijer et al., 2021; Campagnolo et al., 2020; van den Broek et al., 2022; Gebraad et al., 2016). Although inducing yaw misalignment for wake steering will reduce the steering turbine's power, the downstream turbine will experience less wake overlap and therefore a higher rotor-averaged wind speed, resulting in a net increase in power production for the turbine pair. The effectiveness and viability of wake steering have been demonstrated in simulation (Gebraad et al., 2016; Howland et al., 2022a; Howland, 2021; Debusscher et al., 2022), wind tunnel tests (Campagnolo et al., 2020), and field campaigns (Doekemeijer et al., 2021; Howland et al., 2022b; Ahmad et al., 2019). However, tests within utility-scale wind plants have thus far been relatively simple; primarily applied to a small number of turbines, in limited wind conditions, or using simple control algorithms and actuation techniques, e.g., low-dimensional yaw misalignment lookup tables (LUTs) or static offsets applied to wind vane signals. Thus, wind farm control via wake steering remains relatively immature and has not achieved widespread commercial adoption.

More sophisticated, model-based controllers have been tested in simulation and simple experiments and show promise. The engineering wake models (EWMs) these controllers use have the ability to incorporate more dimensions than just wind speed and direction, e.g., turbulence intensity (TI), turbine state, derating, wind shear, wind veer, etc., while remaining sufficiently computationally efficient to be used in real time. "Closed-loop" models also incorporate mechanisms for ongoing tuning to better match observations, thereby enabling more effective wind farm control (Doekemeijer et al., 2020; Howland et al., 2022a), which is important since EWMs omit some physics to remain computationally feasible. Howland et al. (Howland et al., 2022a) in particular showed how a simple LUT-based controller would fail to optimize plant power in all wind condition regimes.

In this study we develop a model-based wake steering controller with a novel hybrid architecture. The model therein uses data-driven input estimation and calibration of an EWM, along with a neural network-based "output corrector". One key feature of this model architecture is that its complexity can be incrementally increased over time, enabling immediate deployment, but allowing for continual improvement as operational data are collected, hence closing the loop. We deploy a minimally-tuned version of this controller to 10 turbines within a 58 turbine utility-scale wind farm, omitting the output corrector in order to collect a validation dataset for model training and to demonstrate the viability of this hybrid approach for closing the loop and improving model predictive capability—and therefore optimizing performance—over time.

## 2 Methods

The primary objective of this study was to develop and deploy a wake steering control system that could be applied to wind plants of 50 or more turbines in potentially complex layouts, with varying numbers of online turbines, power limits, and atmospheric conditions, as would be encountered in the real world. The control problem can be thought of as seeking to maximize the sum of all turbines' power production, which is a function of the wind characteristics (speed, direction, TI, etc.) and turbine states—online status, power limit, nacelle direction or yaw angle, the latter of which is the only means of actuation.

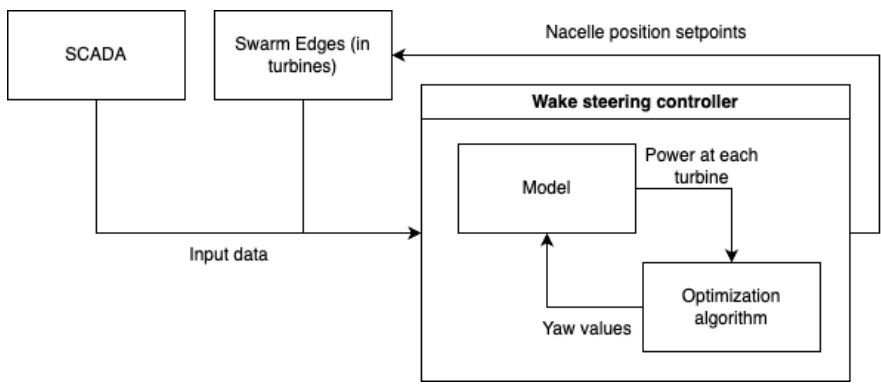

**Figure 1.** Wake steering controller system architecture.

Although instantaneous farm power is straightforward to observe through the SCADA system, there is currently no analytical form available to compute optimal nacelle directions for real-time wake steering control. Despite being shown to be effective in wind tunnel experiments at constant wind direction (Kumar et al., 2023), an extremum-seeking controller would be impractical for the current application given the large number of actuation variables, the variability of wind direction, the relatively long advection times due to the 13D spacing, and the difficulty "dithering" the turbines' yaw position. A LUT-based controller would require many dimensions to capture all the various permutations of wind conditions and turbine states, and would therefore require a prohibitive number of simulations or experiments to develop. For the commercial wind plant studied here, described in Sect. 2.2, a LUT-based controller that allows for variation in wind speed and direction across the plant would contain on the order of 100 billion rows. Thus, a model-based controller was chosen for the present work.

## 2.1 System architecture

The high-level system architecture of the control system is shown in Fig. 1. In this system, a centralized controller is connected to the SCADA system collecting signals (real power, turbine state, rotor speed, etc.) at a nominal 1 Hz frequency from all turbines (Post et al., U. S. Patent, 0 243 699, Aug. 2022). The controller is connected to WindESCo Swarm Edge devices installed in each turbine to be actuated and sends absolute nacelle position setpoints rather than yaw offsets to the turbines, which allows the control problem to be solved in the global coordinate system instead of being relative to the local wind measurement (Post et al., U. S. Patent, 0 272 775, Aug. 2023). This alleviates any issues with yaw measurement nonlinearity (i.e., the measured yaw error at the wind sensor is distorted when the turbine is yawed) and helps the system account for local variation or biases in wind characteristics. In contrast, a non-global control algorithm would make decisions based on a yawing turbine's local wind direction, but the wind direction could be slightly different at a downstream turbine, causing suboptimal setpoint selection. Every minute, an optimization algorithm uses the wake model to predict farm power as a function of yaw to determine the optimum values, and corresponding nacelle position setpoints are subsequently sent to the Swarm Edges.

Digging one level deeper, the wake model architecture is shown in Fig. 2. This model is split into three parts:

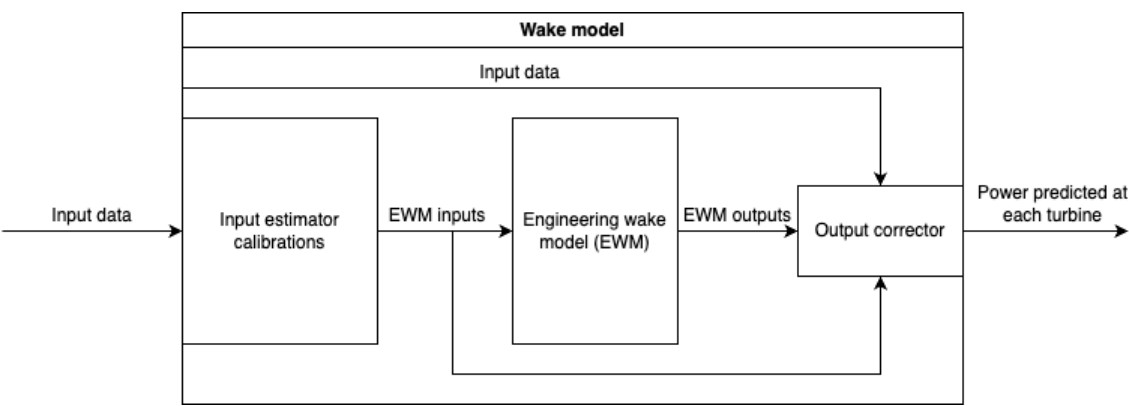

**Figure 2.** Wake model architecture.

1. Calibrated input estimation

2. A physics-based EWM

3. Neural network-based output correction

In the input estimation step, raw data collected from the plant is transformed into input values suitable for the steady-state EWM. The input calibrations use relatively simple models, some of which can be derived from historical data in which wake steering was not active, but these calibration models are critical to getting accurate predictions from the EWM, as we will see later. Lastly, all input data, EWM inputs, and EWM outputs are sent to the output corrector, which has the job of making up for any physics not captured in the EWM. The input estimation and output correction parameters are intended to be solved for in an automated offline process, after which the controller is updated approximately daily. This is in contrast to other closed-loop algorithms, which might update their parameters every iteration.

## 2.2 Pilot installation and data collection campaign

A preliminary version of this wake steering control system was deployed to a commercial onshore wind farm in Utah, USA. The initial deployment did not use an output corrector, as the primary goal was to collect data where wake steering was attempted by a minimally-tuned model to develop the full closed-loop model training and validation process.

The wind farm layout is shown in Fig. 3. This site consists of 58 Clipper Wind C99 model turbines, each with a hub height of 80 meters, a rotor diameter ($D$) of 99 meters, and a rated power of 2.5 MW. The transverse spacing between turbines is approximately $2.8D$, while the downstream turbine row spacing is about $13D$. For the pilot deployment, 10 out of the 58 turbines were augmented with hardware and software to enable wake steering control, with the assumption that the remaining turbines (including turbines at two adjacent co-operated sites; 165 turbines in total) would eventually be instrumented and controlled. To the best of our knowledge, this is the largest campaign of its kind, with the next largest campaign being that

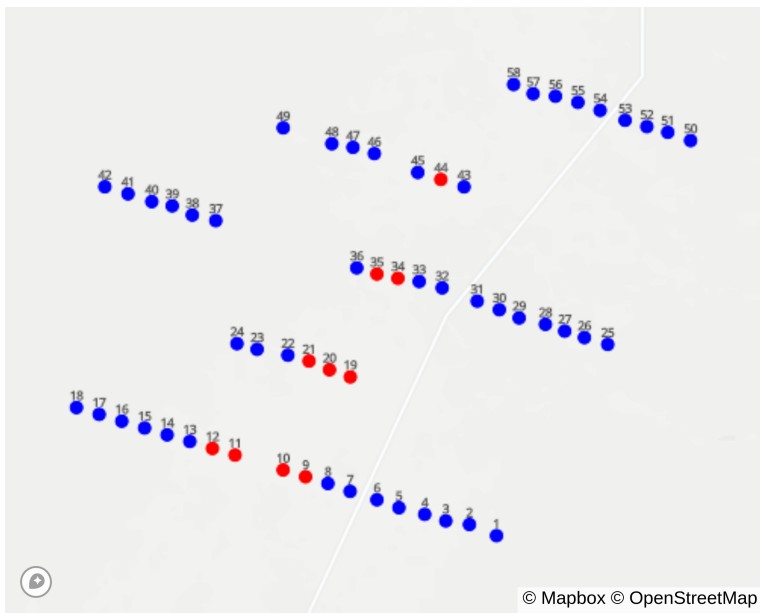

**Figure 3.** The distribution of wind turbines (colored circles) at the wind farm studied here. Red circles indicate turbines with Swarm Edge hardware and software. Turbines IDs are labeled to aid future discussions.

of Howland et al. (2019) where six utility-scale turbines were wake steered. Further, this is the first wake steering control
system deployed that uses absolute nacelle position setpoints sent from a centralized controller.

Waking occurs at a variety of wind directions, but the predominant waking wind direction is from the south and south-
southwest as demonstrated by the wind rose in Fig. 4. For this reason, and because two co-operated wind farms situated
immediately north of the site can cause wind farm waking under northerly wind conditions, we will focus our attention to
southerly winds in this study. The farm layout presents a challenge for wake steering control given its relatively long spacing
in the predominant wind direction and close spacing in the transverse wind direction, meaning that flow control will need to
occur over relatively long distances and with relatively high precision to avoid steering wakes into turbines instead of away
from them. This larger distance might also mean lower wake losses since there is more room for wake spreading and recovery.
As we'll show later, wake losses are approximately 17% when second row turbines are directly waked, thanks to significant
time spent at stable atmospheric conditions.

It should be noted that at this spacing a downstream rotor's half-width represents an angle of approximately 2 degrees,
meaning that a small shift in wind direction can produce a large change in the location of the wake, and therefore its impact on
a downstream turbine. It therefore follows that a controller will need a highly accurate estimate of wind direction to determine
optimal yaw offsets.

The pilot wake steering data collection campaign took place over 4 months, from December 1, 2021 to April 1, 2022. During
this time, we gathered data with wake steering toggled on and off every hour to attempt to collect equal amounts of data in
each state at similar wind conditions. An preliminary minimally-tuned wake model was deployed on the farm during this

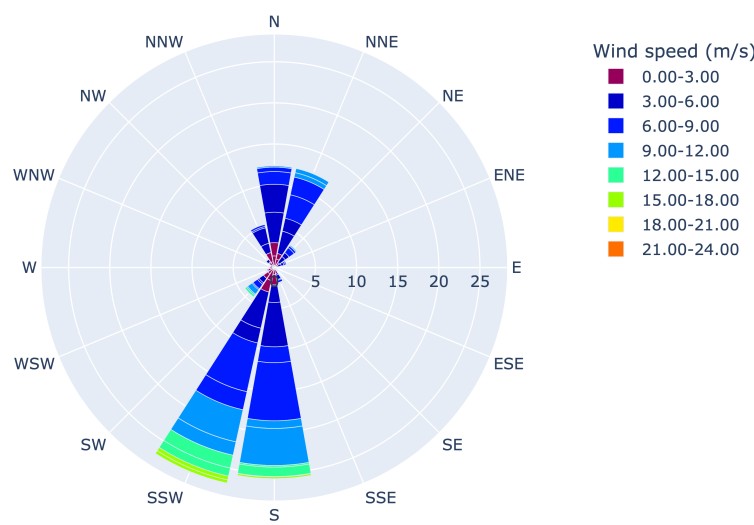

**Figure 4.** The wind rose for the three years prior to the start of the pilot test (September 2018–September 2021).

measurement campaign. As we will see in Sect. 3.1, while this model did not predict farm power with acceptable accuracy, and therefore likely did not achieve optimal wake steering performance, it did show evidence of power gains at downstream turbines when wake steering. More importantly, this pilot data collection phase allowed us to naturally collect training data in a wide range of wind conditions and for various wake steering scenarios, enabling the closed-loop model training and validation process to produce a model that should achieve optimal wake steering performance going forward.

### 2.3 Engineering wake model

NREL's FLOw Redirection and Induction in Steady State (FLORIS) EWM (Version 2.4) was selected for the current work NREL (2021) due to its prevalence within the wind energy community (e.g. van Beek et al., 2021; Meyers et al., 2022), its applicability to wind farm control, and its computational speed and simplicity. FLORIS predicts the steady-state flow field in a wind farm, including the power of each turbine, given the ambient wind conditions and the wind turbine and wind plant configuration information. Typical FLORIS model wind inputs include the ambient wind direction, ambient wind speed, ambient TI, wind shear, and wind veer. Other FLORIS model wind farm inputs include each turbine's geometric information (hub height, rotor diameter, spatial coordinates) and performance characteristics (power curve, thrust curves, and typical power losses under off-yawed conditions).

FLORIS does have a wide range of sub-models to select from, each with their own tuning parameters, e.g., to control how fast wakes recover or deflect when steering. The default sub-models and parameters have typically been determined through

a combination of theory, experiments, and high-fidelity simulations (Doekemeijer et al., 2020; Bastankhah and Porté-Agel, 2016). Since this study is focused on estimating ambient conditions and using machine learning models, we used FLORIS with

135 its default wake models and parameters, which are described in detail in NREL (2021). These included the Crespo–Hernandez wake turbulence model (Crespo et al., 1996), the Sum of Squares Freestream Superposition wake combination model, the Gauss deflection model (Bastankhah and Porté-Agel, 2016; King et al., 2021), and the Gauss–Legacy velocity model (Bastankhah and Porté-Agel, 2016; Niayifar and Porté-Agel, 2016).

## 2.4 Input estimation and calibration

We now discuss the approach for preprocessing data to estimate the input conditions passed to the EWM, along with the model state (i.e., calibrations) to produce the most accurate power predictions possible.

### 2.4.1 Turbine performance curves

To predict turbine power output, FLORIS computes the flow field velocity and then uses the nondimensional power coeffi-
145 cient ($C_P$) to determine the power at each turbine. $C_P$ is defined as

$$C_P = \frac{P}{\frac{1}{2}\rho A U_\infty^3},\tag{1}$$

where $P$ is the power produced by the turbine, $\rho$ is the air density, $A$ is the rotor swept area, and $U_\infty$ is the ambient wind speed.

We consider two possible approaches for determining the power coefficient curves:

1. Use the values as published by the original equipment manufacturer (OEM). This approach was used in our initial
deployment.

2. Infer the power coefficient curves from historical SCADA data.

Although the curves derived using both methods should be similar, it is common for them to differ due to differences between the idealized turbine performance assumed by the OEM and its actual performance on site, or due to changes in nacelle transfer functions. Since we know our wind speed estimation algorithm will use nacelle wind speed, it makes sense then to compute
the $C_P$ curves as a function of the nacelle wind speed. Doing so will implicitly include nacelle transfer function effects on the measured wind speed, removing this potential source of bias when mapping wind speed to power. Note that with this approach it is possible to compute unrealistic power coefficients, i.e., those above the Betz limit, as a result of inaccurate nacelle transfer functions. However, since FLORIS only uses the $C_P$ curve to compute power after computing wind speed, overall, the resulting power predictions should be consistent with the power curve as computed against nacelle anemometer
measurements, and therefore consistent with observed SCADA data.

Figure 5 compares the OEM $C_P$ curve with one modeled using three years of historical SCADA data; differences between the two curves are most evident in the 5 to 12 meters per second range. Turbines that were waked, offline, or derated were

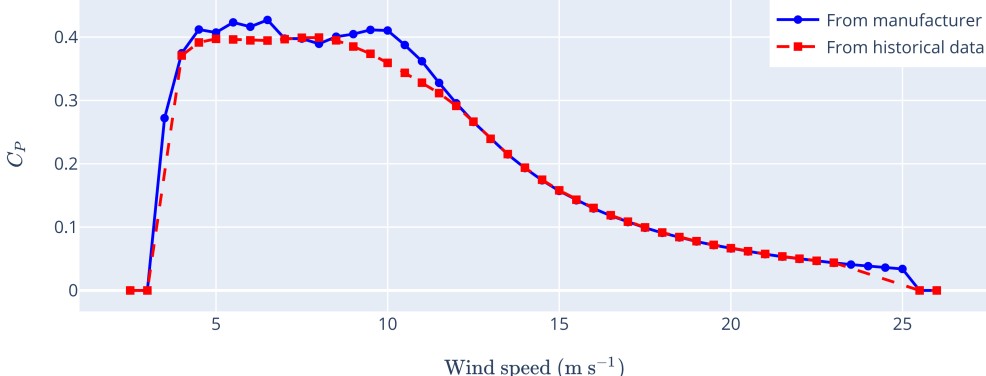

**Figure 5.** The $C_P$ curve from the OEM and derived from historical SCADA data. Note that there is a minor discrepancy at 6.5 m/s in the OEM curve, where our transcription was 553 kW instead of the OEM's 533 kW.

removed from the dataset, as were turbines whose nacelle wind speed was below the OEM cut-in wind speed. The last filtering step was added to avoid biased $C_P$ estimates at low wind speeds, as turbine power is often strongly affected by the rotor inertia. While it is possible to also determine turbine-specific $C_P$ curves using historical SCADA data, this approach resulted in noisy results (particularly for turbines undergoing frequent maintenance or derating). Hence, a single $C_P$ curve was derived and applied to all turbines since all turbines on the farm were of the same model. While in some scenarios (e.g., if measurement biases differ between individual turbines) assuming a single $C_P$–wind speed curve may lead to biases in the model power predictions, this was not observed to be a significant issue on this particular wind farm.

FLORIS also requires thrust coefficient ($C_T$) curves, which govern momentum extraction and affect wake deficits. The thrust coefficient $C_T$ is defined as,

$$C_T = \frac{T}{\frac{1}{2}\rho A U_\infty^2},$$ (2)

where $T$ is the thrust produced by the turbine. Unlike the $C_P$ curves, there is no independent way to verify the $C_T$ curve. Therefore, the values defined by the OEM were used, which implicitly assumes that the wind speeds in the thrust coefficient values match those from the nacelle anemometers.

### 2.4.2 Power loss due to yaw misalignment

The power of a turbine will decrease under non-zero yaw angles. Capturing the proper relationship between yawing and power loss is important due to its influence on wake steering optimization. This relationship is typically modeled through a cosine

exponent $p$,

$$P = P_0 \cos^p \gamma, \tag{3}$$

where $P$ is power produced by a turbine with a yaw error of $\gamma$ and $P_0$ is the power that would be produced at $0°$ yaw error.

The FLORIS default value of $p$ is 1.88, which is based on LES results (Gebraad et al., 2016). (To model power production properly above rated wind speeds, FLORIS adjusts the turbine's effective wind speed rather than the power using a cosine exponent of $p/3$. This effective wind speed is then used to look up the power.) Subsequent studies have shown that $p$ may vary across different turbines and wind conditions (Howland et al., 2020; Liew et al., 2020). However, we did not observe a significant difference in model performance when using the default value of $p$ versus values obtained through a separate estimation procedure based on SCADA data. Therefore, the FLORIS default $p$ value of $1.88$ was used in this study. Although we use the default cosine exponent, the power loss with yawing relationship will be implicitly accounted for in our output corrector model discussed later.

### 2.4.3 Ambient wind speed estimation

One required input to the FLORIS wake model is the ambient wind speed. First, we apply a rolling average in time on the high-frequency wind speed measurements, i.e.,

$$\bar{U}_i(t) \equiv \frac{1}{T_U} \int_{t-T_U}^{t} U_i(t')\, dt', \tag{4}$$

where $\bar{U}_i(t)$ is the time-filtered velocity of turbine $i$ at time $t$, $U_i(t)$ is the high-frequency velocity of turbine $i$ at time $t$, and $T_U$ is the size of the averaging window (the subscript $U$ indicates that it applies to the velocity). We found model performance to be generally equivalent for time filters between 1 and 10 minutes, and therefore used a constant value of $T_U = 5$ minutes in this study.

We next need to determine which of these turbine velocities correspond to unwaked turbines, since the wind speed inputs to our waked model correspond to ambient wind conditions. We estimate which turbines are unwaked based on the estimated ambient wind direction computed in Sect. 2.4.4 and the geometric parameterization in IEC 61400-12-1 (2) (see Fig. A.1 of this reference). From our experimental data, we observed that under periods of high atmospheric stability, wakes can persist for longer distances than the 20 rotor diameter cutoff assumed in IEC 61400-12-1 (2), and therefore we require a turbine to be at least 30 rotor diameters downstream from a potential waking turbine to be considered unwaked.

Since wind speeds measured at operational and non-operational turbines may be different (e.g., nacelle anemometer wind speeds are affected by rotor-induced flows and nacelle transfer functions) and the $C_P$ curve in Sect. 2.4.1 was derived only from operational turbines, it is recommended to exclude wind speed measurements from non-operational turbines. This can be done by excluding turbines whose power or rotor speed falls below some threshold. Even after these filtering steps, however, it is possible that a turbine will exhibit implausible nacelle anemometer wind speed measurements. These can be excluded by filtering any turbine wind speeds that are a certain number of standard deviations from the mean value over all turbines, though this treatment was not found to be necessary for wind speed in the current study.

We finally apply a spatial filtering step to translate turbine-level time-filtered wind speed measurements to wind speed inputs in our wake model. That is, for turbine $i$, we compute the ambient, spatially averaged wind speed $U_{\infty,i}(t)$ as

$$U_{\infty,i}(t) \equiv \frac{\sum_j w(\mathbf{x}_i, \mathbf{x}_j)\bar{U}_j(t)}{\sum_j w(\mathbf{x}_i, \mathbf{x}_j)}, \tag{5}$$

where the summation $\sum_j$ takes place over all turbines (but excluding those removed from the analysis due to waking, offline states, or implausible data), with weights between a turbines $i$ and $j$ with Cartesian positions of $\mathbf{x}_i$ and $\mathbf{x}_j$ defined as

$$w(\mathbf{x}_i, \mathbf{x}_j) \equiv \frac{\exp\left(-r^2/2\right)}{\sqrt{2\pi}}, \tag{6}$$

where

$$r \equiv \frac{\|\mathbf{x}_i - \mathbf{x}_j\|/D}{\mathcal{N}_U}. \tag{7}$$

$D$ denotes the turbine rotor diameter, and $\mathcal{N}_U$ is a non-dimensional scaling factor corresponding to the number of rotor diameters to use in the spatial filter. Small values of $\mathcal{N}_U$ allow a larger degree of wind speed heterogeneity or non-uniformity in the model, whereas high values of $\mathcal{N}_U$ correspond to a more homogeneous wind speed field.

$\mathcal{N}_U \to \infty$ is equivalent to a farm-wide average wind speed, which is the most common approach in the literature (Doekemeijer et al., 2020; Howland et al., 2022b). For large farms, however, it may be important to capture spatial wind speed variations to accurately predict power levels. Doekemeijer et al. (2022) observed heterogeneous wind speeds in the industrial-scale wind farms in their study, but also noted difficulties in accurately modeling these heterogeneous wind speeds with FLORIS, due to either deficiencies in the FLORIS model or uncertainties in the background wind conditions. They derived a generalized inflow profile from annual wind speed measurements and used this profile to translate a homogeneous farm-averaged wind speed into heterogeneous turbine-level wind speeds. The advantage of our approach over Doekemeijer et al. (2022) is that it can account for more instantaneous spatial variability in the wind speed on a farm. We will assess the effect of including wind speed heterogeneity in our model in Sect. 3.

### 2.4.4 Ambient wind direction estimation

The approach for ambient wind direction estimation mirrors much of that for ambient wind speed estimation, but with the additional complexity associated with the use circular quantities.

As before, we start by applying a time filter to $\theta_i(t)$, the high-frequency wind direction signal for turbine $i$:

$$\bar{\theta}_i(t) \equiv \mathrm{atan2}\left(\frac{1}{T_\theta}\int_{t-T_\theta}^{t}\sin\theta_i(t')\,dt', \ \frac{1}{T_\theta}\int_{t-T_\theta}^{t}\cos\theta_i(t')\,dt'\right), \tag{8}$$

where $\mathrm{atan2}$ is the two-argument arctangent function, and $T_\theta$ is the wind direction temporal filter width. (As with the wind speed (Sect. 2.4.3), we observed little dependence on this parameter for $T_\theta$ between 1 and 10 minutes and therefore used a

constant value of $T_\theta = 5$ minutes.) Outlier filtering is then applied to $\bar{\theta}_i(t)$ to exclude offline turbines and turbines whose estimated wind directions differ significantly from the mean wind direction (in a circular sense) across the farm.

At this stage, as with the wind speeds, we apply a spatial filtering process (with nondimensional wind direction filter width $\mathcal{N}_\theta$, analogous to wind direction filter width $\mathcal{N}_U$ from Sect. 2.4.3) to obtain the spatially and temporally filtered ambient wind direction $\theta_{\infty,i}(t)$. For estimating wind direction, the nearly universal approach in the literature (Fleming et al., 2017; Ahmad et al., 2019; Howland et al., 2019; Fleming et al., 2020; Simley et al., 2021; Doekemeijer et al., 2021, 2022) is to use a single wind direction (corresponding to $\mathcal{N}_\theta \to \infty$). In our initial pilot study (Sect. 2.2), however, we used $\mathcal{N}_\theta = 5$ to attempt to model wind direction heterogeneity on the farm. We determined, however, that this did not improve the model performance on this farm, presumably due to difficulties in FLORIS of accurately modeling heterogeneous wind directions. For that reason, we switched to a simple farm average for the wind direction in the remaining models studied in this paper (see Sect. 3.2). We denote this farm-averaged wind direction as $\theta_\infty(t)$.

When computing the wind directions, there are two potential sources of the high-frequency wind direction $\theta_i(t)$. The simplest is the SCADA wind direction signal; however, this signal is often not calibrated to true north and even after calibration, may not be sufficiently accurate to use for the purposes of wake steering. We also have access to the nacelle direction reported by GNSS compasses in the Swarm Edges installed on 10 turbines. The wind direction can be computed from the GNSS compass nacelle direction and the measured yaw error, and we expect this measure of wind direction to provide a higher level of accuracy. In our initial "naive" model deployment, we used a mix of both signals: GNSS compass wind directions were used where available, and SCADA nacelle directions (that were manually calibrated in the past based on direction readings taken by operators in the field) were used otherwise. However, this approach gave unrealistic wind direction predictions (due to issues calibrating the SCADA wind directions and inconsistencies in the sensors), and so we switched to using only GNSS compass wind directions in all other models, assigning turbines without GNSS compasses zero weight in the spatial filtering process.

The wind direction obtained through this approach, however, may differ from that needed in the model to best capture waking, either due to sensor biases, underlying atmospheric conditions, or model inaccuracies. One approach for tuning the wind direction to best capture waking is to use the input measurements as is but to adjust model coefficients (e.g., those pertaining to veer or wind direction deflection by the rotor). This approach is not preferred, however, in the case of sensor biases, since we may be tuning model coefficients to unrealistic values to compensate for inaccurate input conditions. We therefore instead adjust the wind directions themselves following a similar approach to that outlined in Kanev (2020) and Doekemeijer et al. (2022).

Figure 6 shows the target and reference turbines selected for estimating the wind direction adjustments. Target turbines are subjected to waking from one row of upstream turbines when wind is from the south. Reference turbines are the southernmost turbines, excluding turbines outfitted with Swarm Edges to avoid confounding effects on power from high yaw errors. We could extend our analysis to deeper turbine rows to determine offsets for these turbines. However, we observed that the waking signal became weaker and more diffuse for these turbines, making parameter estimation difficult. A similar conclusion was reached in (Kanev, 2020).

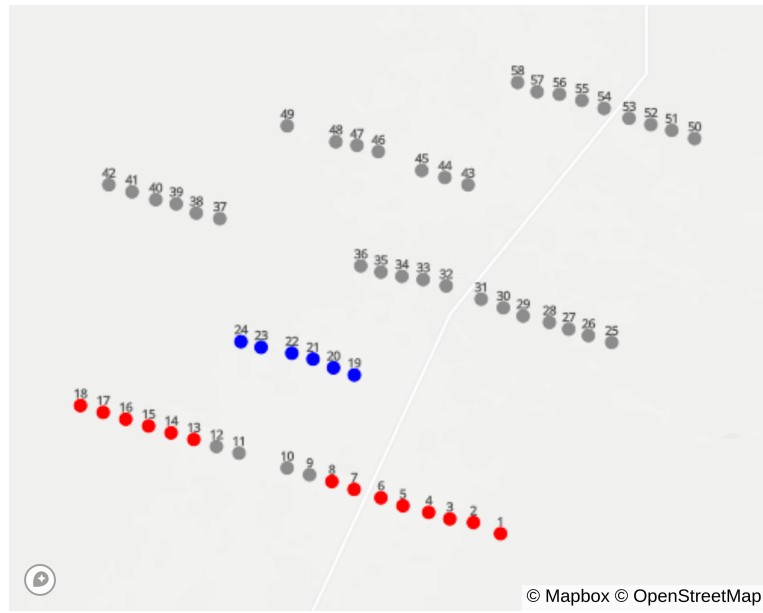

© Mapbox © OpenStreetMap

**Figure 6.** Turbines selected as targets (blue) and references (red) for estimating wind direction offsets.

For each target turbine $i$, we compute the power ratio at time $t$, $R_i(t)$, as

$$R_i(t) = \frac{P_i(t)}{\frac{1}{N_{ref}} \sum_r P_r(t)}, \tag{9}$$

where $P_i(t)$ is the power produced by turbine $i$ at time $t$, $N_{ref}$ denotes the number of reference turbines selected, and $\sum_r$ indicates a summation over each reference turbine. Following the typical convention in the literature (Kanev, 2020; Fleming et al., 2020; Doekemeijer et al., 2022), these temporal results are then binned by the farm-averaged ambient wind direction $\theta_\infty(t)$, and the mean power ratio in each bin is reported. Results are computed over four months of operational data using a five-minute temporal averaging period. Prior to binning, we filter target and reference turbines for yaw errors below $5°$ (to

avoid power losses due to yawing), region 2 wind and power conditions, and wind directions between $152°$ and $242°$ (measured clockwise, with $0°$ indicating true north; to ensure wind is from the south).

For a given wind direction bin, we compute the model predictions of the power ratio as a function of wind direction by running FLORIS with a single wind direction (corresponding to the center of the bin), wind speed (corresponding to the overall mean wind speed in the data set), and a TI of 10%. While we could obtain more representative model results by simulating

each of the wind directions, wind speeds, and turbulence intensities in our observations, the primary focus of our analysis at this stage is in matching the directional pattern of waking and unwaking in the data, for which a fixed wind speed and TI is sufficient and computationally advantageous.

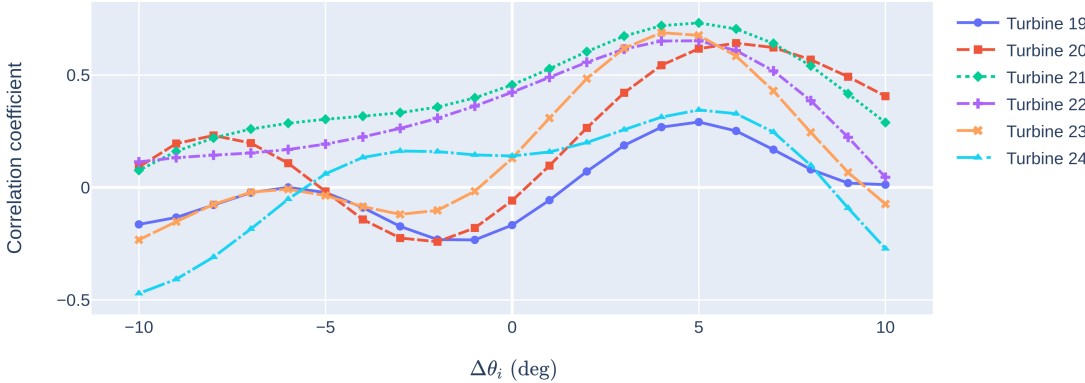

**Figure 7.** Correlation coefficients computed at various wind direction offsets for each of the target turbines. Turbines are numbered 19–24 corresponding to their identification in Fig. 6.

Following Doekemeijer et al. (2022), we determine the optimal offset to the wind direction $\Delta\theta_i$ at each turbine to maximize the correlation between the binned power ratios in the observations $R_i(\theta_\infty)$ and those from the model $R_i^{\text{model}}(\theta_\infty)$:

$$\Delta\theta_i \equiv \arg_{\Delta\theta_i} \max \left( r\left[ R_i(\theta), R_i^{\text{model}}(\theta + \Delta\theta_i) \right] \right). \tag{10}$$

Here, $r$ is the Pearson correlation coefficient. We selected the correlation coefficient as our optimization function rather than a quantity like the mean-squared error because our model has not yet been tuned to match the waking strength observed in the data. At this stage, we simply want to ensure we can match the directional pattern of waking and unwaking.

Figure 7 shows the correlation coefficients for each of the 6 target turbines identified in Fig. 6. Wind direction offsets ranging from $-10°$ to $10°$ are used under the assumption that the wind direction shift in excess of $10°$ is not needed. (Negative values indicate a counter-clockwise correction to the wake direction, whereas positive values indicate a clockwise correction.) If no wind direction offset were needed, the correlation coefficients should peak at $\Delta\theta_i = 0°$. The fact that the peak is offset from $0°$ indicates that this additional adjustment to the wind direction signal is necessary. For many of the turbines, we observe two nearly identical peaks: one near $-7°$ and one near $+5°$. This pattern makes sense geometrically: because of the close row-wise spacing of the front-row turbines, many of the target turbines (19, 20, 23, 24) can be waked by a different front-row turbine as a result of a $12°$ ($-7°$ to $+5°$) wind direction shift.

The challenge is then in determining whether a turbine should have a positive or a negative wind direction shift. To resolve this, we consider turbine 22. Because there is a gap in the front-row turbines upstream of turbine 22 (see Fig. 6), we do not expect the same symmetrical pattern in the correlation coefficients. Indeed, we notice that turbine 22 exhibits a peak for $\Delta\theta_i = 5°$ but no corresponding peak for negative $\Delta\theta_i$. Figure 8 compares the mean power ratio in the data to that predicted

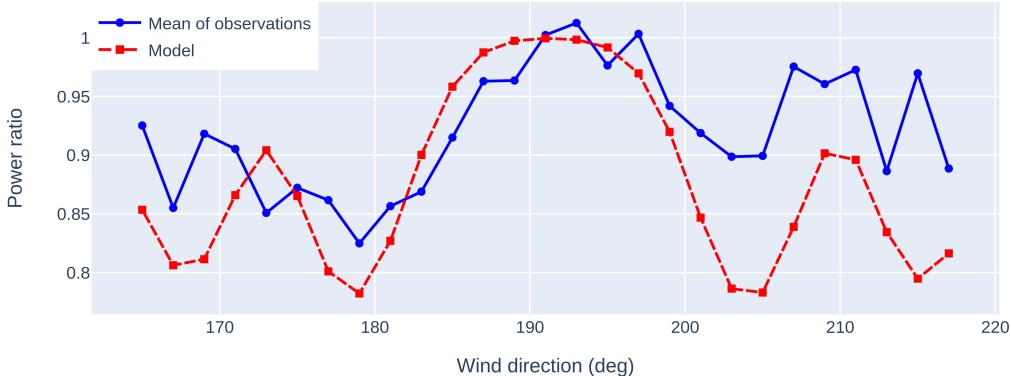

**Figure 8.** Comparison between model predictions and mean of observations for the power ratio of turbine 22 as a function of wind direction under a $+5°$ wind direction shift in the model.

by the model under a $+5°$ wind direction shift. The agreement in trends between the data and model is good, showing limited waking (power ratios near 1) for wind directions around $190°$ (corresponding to the gap in the front-row turbines) and waking (power ratios near 0.8) around $180°$ and $200°$.

Based on this analysis, we assign a positive wind direction offset of $5°$ for turbine 22 and assume the positive solution branch also holds for the remaining turbines. Since most target turbines show a peak at $+5°$ (turbines 20 and 23 are slight exceptions, with peaks at $+6°$ and $+4°$, respectively), we apply a single, global $+5°$ wind direction offset.

To explore the sensitivity of the wind direction offset to the time range considered, we repeated this analysis over different 2-month training periods within the overall dataset. While there were small variations in the optimal wind direction offsets between different times for certain turbines (typically on the order of $1°$–$2°$), a farm-wide wind direction offset of about $5°$ was a consistent finding. We therefore use a global, time-invariant offset in wind direction.

### 2.4.5 Ambient turbulence intensity estimation

The ambient TI computation is performed similarly to the ambient wind speed computation presented in Sect. 2.4.3, with an additional preliminary step of computing the TI $I_i(t)$ from the high-frequency wind speed:

$$I_i(t) \equiv \frac{\sigma_{U_i(t)}}{\langle U_i(t) \rangle}, \tag{11}$$

where $\sigma_{U_i(t)}$ denotes the standard deviation of the high-frequency wind speed for turbine $i$ over the previous 10-minute time window, and $\langle U_i(t) \rangle$ is the mean of the high-frequency wind speed of turbine $i$ over the same time window.

The approach then closely parallels the wind direction approach, with removal of TI values from waked turbines and optional outlier filtering. We finally apply a simple spatial average to obtain a single turbulence intensity value $I_\infty(t)$ to use in the model at time $t$.

The estimate of the ambient TI $I_\infty(t)$ made directly from nacelle anemometry is not necessarily the best input to the FLORIS model. When modeling TI, nacelle anemometers, by virtue of their location downstream of the rotor, are affected by rotor-induced turbulence and so do not capture the ambient flow characteristics (Smith et al., 2002). Wind turbine manufacturers introduce a nacelle transfer function to estimate the ambient wind speed based on measurements behind the rotor; however, the theory, implementation, limitations, and accuracy of a given nacelle transfer function are rarely disclosed. For example, St. Martin et al. (2017) found that different nacelle transfer functions should be used based upon the atmospheric stability and turbulence levels. In addition, even if we had perfect knowledge of the ambient turbulence intensity, this quantity will likely not exactly correspond to the FLORIS model TI, which should be viewed as a model parameter that influences wake spreading and stability rather than as a perfect correlate of the ambient TI observed in the field.

For these reasons, we considered an additional step to map the ambient TI as measured by the nacelle anemometer ($I_\infty$) to that needed in FLORIS model ($\mathcal{I}_\infty$) to better resolve the observed waking conditions. To do so, we first select turbines, wind directions, and wind speeds with high levels of waking (as indicated by frequent occurrences of low power ratios) and denote these as waking cases of interest. Historical data corresponding to each waking case of interest is then binned by the ambient nacelle TI. In each nacelle TI bin, we compute the optimal turbulence intensity to use in FLORIS to minimize the mean-squared difference between the power ratio of the turbine in the historical data for a given wind speed and direction, $R_i(I_\infty \mid U_{\infty,i}, \theta_\infty)$, and that predicted by the model, $R_i^{\mathrm{model}}(\mathcal{I}_\infty \mid U_{\infty,i}, \theta_\infty)$:

$$\mathcal{I}_\infty(I_\infty) \equiv \arg_{\mathcal{I}_\infty} \min \left( \left[ R_i(I_\infty \mid U_{\infty,i}, \theta_{\infty_i}) - R_i^{\mathrm{model}}(\mathcal{I}_\infty \mid U_{\infty,i}, \theta_{\infty_i}) \right]^2 \right). \tag{12}$$

Equation (12) gives optimal FLORIS turbulence intensity values $\mathcal{I}_\infty$ for a range of different nacelle turbulence intensities $I_\infty$, from which a regression relationship can be formulated.

In the initially deployed model (see Sect. 2.2), we determined the following TI mapping based on a single waking case of interest:

$$\mathcal{I}_\infty = 0.0571 - 1.83 I_\infty + 29.6 I_\infty^2 - 67.1 I_\infty^3. \tag{13}$$

However, we found that while a given mapping tended to improve the power predictions of the turbine under consideration, the results did not generalize to other turbines in the wind farm. The model without a TI mapping gave the most consistent results over the range of turbines considered, and so we chose to omit the TI mapping for the new models developed here. We expect, however, that TI mapping may still be beneficial on other farms, especially those where the nacelle anemometer TI reading differs significantly from the ambient TI.

## 2.5 Output corrector

The output corrector architecture devised is shown is Fig. 9. Available inputs are the raw, high-frequency input data from the SCADA system and Swarm Edges, the EWM inputs, and the EWM outputs. The last block before returning predicted power

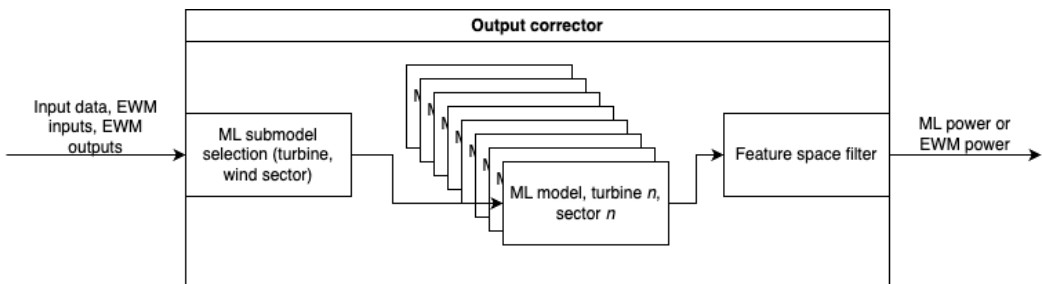

**Figure 9.** Output corrector architecture.

is a so-called "feature space filter", which prevents the output corrector from returning its machine learning predictions for feature values (in a multidimensional sense) very far from those upon which it has been trained, falling back to the EWM predictions in these cases. Again, the purpose of the output corrector is to provide the most accurate predictions possible for the power of every turbine on the plant. The output corrector also must be able to handle varying yaw angles for any upstream turbines to be used for optimization.

In general, the measurement system provides signals at turbine $i$ at time $t$, representing the power, wind speed, air density, yaw misalignment, and nacelle position for the set of $m$ turbines in the farm $i \in 1, \ldots, m \equiv \mathcal{T}$. In this setup, we wish to develop a set of models that will predict the power at every turbine within the farm. Waked and free-stream turbines are treated differently. The set of free-stream turbines is defined as $\mathcal{T}_f \subseteq \mathcal{T}$, and the set of waked turbines is defined as $\mathcal{T}_w \subseteq \mathcal{T}$, such that $\mathcal{T}_w \cap \mathcal{T}_f = \emptyset$. These signals are combined into a feature matrix in the form $X_{i,s}(t)$, where the subscript $i$ indicates the turbine ID, $s$ indicates the signal name, and $t$ indicates the time stamp. The feature matrix can include raw input data, EWM inputs, EWM outputs, and time-lagged versions of each.

We use this feature matrix as the input into an arbitrary machine learning model framework. The aim is to predict the power at each turbine in the farm at a desired frequency, and at least one model will be trained for each target turbine. In this framework, we have $mTk$ possible features in the full dataset, where $k$ is the number of signals collected, and $T$ in this context describes the number of time lags available for use. In our study, we limit $T$ to 2: the current time and the one minute time lag.. Limiting the discussion to physically relevant signals listed above, including a 1 minute time lag of each signal, and excluding the wind speed and power signals of the target turbine, we have 574 features to choose from when predicting the power of a single turbine.

We can treat the power predicted at a single turbine, $\bar{P}_i(t)$, as a function of the data collected by all other turbines on the farm, $\bar{P}_i(t) = \mathbf{f}(X_{i,s}(t))$. However, this level of dependence creates problems for a machine learning model when a turbine is offline, derated, or disconnected from the SCADA server. Automated imputation methods are challenging to deploy due to the uncertainty in the case of a missing or altered signal. One way to combat this problem is through the use of dimensionality reduction tools, which are commonplace in data science applications.

We use a physics-informed dimensionality reduction to select features with causative relationships between control variables and power production. This algorithm is based on a parameterized modification to (IEC 61400-12-1, 2), which is used to

approximately define the wind direction sector where a turbine is waked by another turbine. The width of the waked sector at turbine $i$ and generated by turbine $j$ is defined by

$$\theta_w[i,j] = \arctan(2.5D_e/L_e + 0.15) + 10 \tag{14}$$

where $D_e$ is the equivalent rotor diameter and $L_e$ is the actual straight line distance between turbines $i$ and $j$ (IEC 61400-12-1, 2). The wind directions where a turbine $i$ waked by turbine $j$ are then computed as $\theta_{min}[i,j] = \theta_c[i,j] + \alpha\theta_w/2$ and $\theta_{max}[i,j] = \theta_c[i,j] - \alpha\theta_w/2$, where $\theta_{min}[i,j]$ and $\theta_{max}[i,j]$ are the minimum and maximum wind directions where turbine $i$ is waked by turbine $j$. $\theta_c$ is the angle between turbines $i$ and $j$ Our modification to this approach is to use a parameterized scaling factor, $\alpha$, and a distance threshold parameter, $D_{\max}$, when computing the wind sector bounds. $\alpha$ controls the final width of the wake sector. $D_{\max}$ is used to discard any waked sectors where $L_e > D_{\max}$. The IEC standard can be recovered using $\alpha = 1$ and $D_{\max} = 20$.

This partitioning algorithm takes the following steps, for each turbine $i$:

1. Determine the set of turbines that wake turbine $i$, $\mathcal{T}_u$.

2. Determine the bounds of each wake sector $\theta_w[i,j] \forall j \in \mathcal{T}_u$.

3. Merge any overlapping wind sectors, noting which turbines generated the wake in that sector.

4. Find all wind directions where turbine $i$ is not waked by any upstream turbine, and add these to the set of wind sectors.

This algorithm is repeated for each turbine on the farm. It results in a set of wind sectors for each turbine, $\Theta_i$. In each wind sector, we have a set of turbines, $\mathcal{T}_{i,\Theta}$ whose signals are relevant to predicting the power at turbine $i$. Subsequently, the model training process is described as follows:

1. For each turbine, $i$, select the set of wind sectors $\Theta_i$.

2. For each wind sector in $\Theta_i$, select relevant turbine signals, $\hat{s}$, from all turbines in $\mathcal{T}_{i,\Theta}$.

3. Remove turbine $i$'s power and wind speed signals from the feature set.

4. Subset the full dataset $X_{i,s}(t)$ to the relevant signals, $X_{i,\hat{s}}(t)$.

5. Find the set of timestamps, $\hat{t}$, where the farm averaged wind direction lies in the chosen wind sector.

6. Train the model on $X_{i,\hat{s}}(\hat{t})$.

Thus, a single model is trained for each turbine and each wind sector. The number of wind sectors varies per turbine and depends on the farm layout. The number of features used to train each model depends on the number of waking turbines in the chosen wind sector.

For the remainder of this study, models for turbines in waked sectors use the following features measured at upstream turbines to predict the power of the waked turbine: Wind direction as measured by the GNSS compass and nacelle anemometer, 1 minute

rolling average of SCADA wind speed, 1 minute rolling average of estimated yaw error, 10 minute rolling TI measured at the nacelle, 1 minute rolling average of SCADA power, 1 minute rolling average of SCADA power lagged by one minute, and the EWM-predicted power. In addition, the models use the same features measured at the target turbine, but with the target turbine's current 1 minute rolling average of wind speed and power measurements omitted (the time-lagged measurements are still used as features.)

For the remainder of this study, models for turbines in free-stream sectors use the following features measured at the turbine of interest to predict its own power: 1 minute rolling average of estimated yaw error, 10 minute rolling TI measured at the nacelle, wind direction as measured by the GNSS compass and nacelle anemometer, 1 minute rolling average of SCADA wind speed, 1 minute rolling average of SCADA power lagged by one minute, and the EWM-predicted power.

Six features are always used to predict a free stream turbine's power. For a waked turbine, $7n + 5$ features are used, where $n$ is the number of upstream turbines used in the modeling. This results in a minimum of 12 features and a maximum of 404, depending on how the partitioning is done.

The default IEC distance cutoff of 20 rotor diameters was used, and the disturbed wind sector width was reduced by a factor of 8 from the IEC value (IEC 61400-12-1, 2). This was then used to partition the input space with a varied number of features per model. A minimum of 1000 samples were required in each sector before a model was trained. A multilayer perceptron was trained for each turbine/wind sector with hidden layers of [20, 50, 50, 50, 50, 20] nodes and a rectified linear unit (ReLU) activation function. Scikit-learn's `MLPRegressor` implementation (Pedregosa et al., 2011) was used in this study.

ReLU functions are piecewise continuous and not differentiable, which can be challenging for gradient-based optimization of farm power. Differentiable activation functions are available in other machine learning libraries, e.g., TensorFlow, though these significantly increase the cost of training and prediction. ReLU suffices for the purposes of this study, since the focus is on predicting observed behavior.

For the feature space filter, a one-class support vector machine was used to identify extrapolation from the training data prior to making predictions. This can be thought of as detecting outliers in new input features compared to the features used to train the model. Any feature that was missing more than 60% of the data was dropped from the feature space during training and prediction. Individual data points (i.e., specific timestamps) were dropped if any remaining features were missing values. Derated and offline data was removed from the training dataset but remained in the test dataset. In the partitioning scheme, these features are dropped on a per-model basis.

There are many potential causes for missing data in the SCADA pipeline, and it is difficult to automatically determine what imputation strategy is most appropriate for any particular missing data point. As a result, no imputation is performed during training or prediction. During the prediction stage, all features used in training must be available for a prediction to be made. Any features that were dropped during training must be dropped during prediction. Because of the frequency with which missing data occurs in the SCADA pipeline, using a large number of features can dramatically reduce the dataset size. By partitioning the farm into small sectors, the waked turbine model is only allowed to depend on a very small number of freestream turbine's signals. This means that the probability that a row has a feature with a missing value is much smaller than if we were to use larger partitions, and this inrceases the dataset size available for training.

## 2.6 Validation methodology

In order to close the loop on the model, a validation process was developed to quantify its fitness for use in a controller. Given the need to predict the farm power accurately, one simple validation metric might be some aggregate of the power prediction error. However, it is hard to know what threshold makes for an acceptable model simply by computing this metric over a given dataset.

We therefore took a phenomenological approach to validation, developing metrics and visualizations that test the model's ability to predict the important phenomena, namely the ability to predict:

– The effects of waking on turbine power—when it occurs and how severely

– The response of the waked power loss to upstream steering

– The power loss incurred by an upstream steering turbine

The model must be able to respond to slow transients in the wind conditions and thus must accurately represent when a turbine transitions between waked and non-waked states. Power gains occur over longer time scales, and thus the model should accurately capture the mean power loss due to waking as a function of wind direction, wind speed, and TI. If a model can predict all of these phenomena accurately on a 1 minute basis, we assume it will optimize the turbine nacelle positions for wake steering control effectively. Results are presented and discussed in the following sections from this perspective.

## 3 Results

### 3.1 Validation results from initial model

We first present some validation results for the initially deployed "naive" or minimally tuned model to motivate the creation and validation of new variants. As discussed in Sect. 2.6, we seek to model waking and unwaking events dynamically and in aggregate. We inspect our dynamic modeling capabilities through an example time series, and then study aggregate modeling through power ratio versus wind direction plots.

One such time series is shown in Fig. 10, which corresponds to a time when wake steering control was enabled. Figure 10(a) shows the estimated wind direction of a downstream waked turbine (turbine 23, blue) and an upstream wake steering turbine (turbine 11, red). (Refer to Fig. 3 for more details on the turbine numbering and layout.) The red horizontal line in this plot denotes the expected wind direction around which the downstream turbine will be waked by the upstream turbine. In Fig. 10(a), we observe a difference in the modeled wind directions of the downstream and upstream turbines, which will complicate our ability to wake steer correctly, since the model needs to account for wake propagation under heterogeneous wind directions. Nevertheless, it is clear that the wind direction is near a direction where the downstream turbine might be waked. Figure 10(b) shows the yaw error of the upstream turbine. From the high yaw errors of the upstream turbine, we deduce that wake steering is occurring throughout much of this time series. (The yaw error of the downstream turbine remains near $0°$, indicating that

it is not actively wake steering, as expected.) Figure 10(c) shows SCADA power and model power predictions for these two turbines. We observe that while the model often predicts minimal waking (e.g., at time 4:05, model power predictions of upstream and downstream turbines are similar, suggesting successful wake steering), the actual data tells a different story: the SCADA power value of the downstream turbine is sometimes well below the upstream turbine (e.g., at time 4:05), indicating strong waking is still present and that our wake steering approach was not successful. Finally, we see that even for the upstream turbine, there are significant differences between the SCADA and model power values. These discrepancies highlight the shortcomings of the initial model and the need for model tuning to perform successful wake steering.

We use power ratio versus wind direction plots (see Sect. 2.4.4) to assess our ability to capture waking and unwaking in aggregate. Reference turbines for this calculation are determined based upon the modified IEC criteria introduced in Sect. 2.4.3. Figure 11 shows the power ratio versus wind direction for turbine 23 (the downstream turbine discussed in the time series plot above), computed over the entire 4 months of our experimental campaign. The data is filtered for wind speeds between 5 and 12 meters per second and powers between 200 and 2500 kW (to ensure region 2 operation). To clarify trends in the data, we use the overlapping binning approach suggested in Fleming et al. (2019), with a bin width of $4°$ and a step of $2°$.

SCADA data and model predictions are shown both when wake steering control is deactivated and when wake steering control is activated. The model and the SCADA show low power ratio values near wind directions of $195°$ and $210°$, indicating that waking is occurring here. The strength of waking in the absence of wake steering, however, is significantly over-predicted by the model (much lower power ratios near waking wind directions in the model than in the SCADA data). From the SCADA data, we observe an increase in the power ratios when wake steering control is activated, demonstrating that wake steering is successfully increasing the power of this downstream turbine. The differences in power ratios with and without wake steering, however, are much larger in the model than in the SCADA data. This indicates that despite the gains, the model does not properly "understand" the plant behavior, and therefore is not able optimize the plant to its full potential, hence the effort here to improve model predictive capability.

## 3.2 New models tested

Four additional models were created to be validated against the pilot dataset. These have increasing levels of complexity to demonstrate the value of each component added to the model and the sensitivity to various model parameters.

The selected models are summarized in Table 1. Model 0 is the initial naive model that was deployed during our pilot campaign. Model 1 represents the baseline model for our model tuning study, with no wind direction offsets, farm-averaging of the wind speed ($\mathcal{N}_U \to \infty$), and no output corrector. We then add the estimated wind direction offset in Model 2 to study its effect in isolation. Model 3 is similar to Model 2, but uses a 10-diameter Gaussian consensus wind speed ($\mathcal{N}_U = 10$) rather than a farm average to study the effect of wind speed heterogeneity. Finally, we use the full output corrector architecture as Model 4. Because neural networks have a high risk of overfitting, Model 4 is a combination of two trained models: one trained on data from December 1, 2021–February 1, 2022, and one from February 1, 2022–April 1, 2022. Each model is used to predict the data held out from its training data. We concatenate the two holdout sets to approximate the aggregate performance of the Model 4 over the entire time range. It is important to note that this is a conservative test of the output corrector capability, since

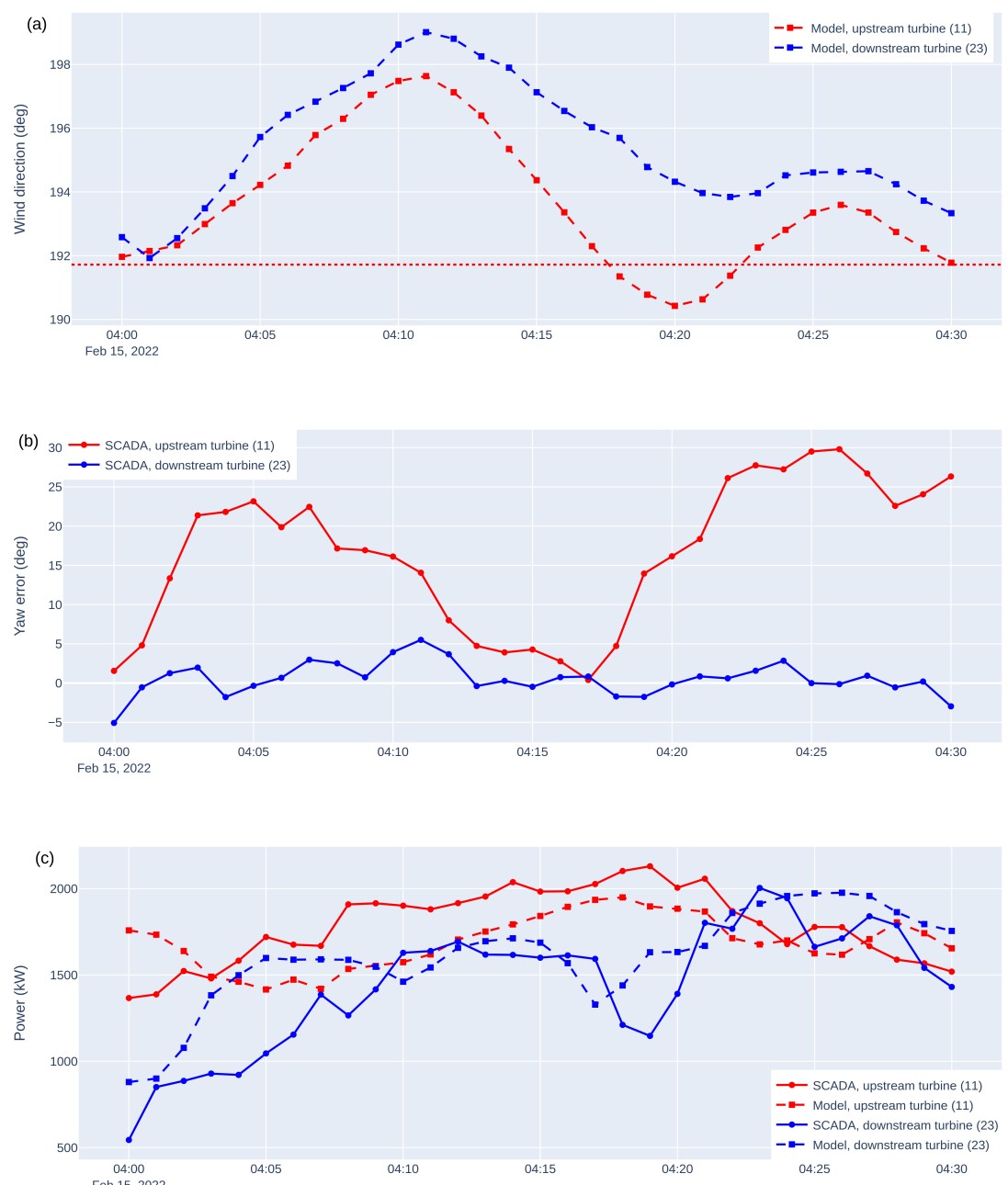

**Figure 10.** Example time series showing wind direction (a), yaw error (b), and power (c) for an upstream turbine (red) and a downstream turbine (blue). Dashed lines indicate model predictions, and solid lines indicate SCADA results. The horizontal dotted line in (a) shows a wind direction where the downstream turbine (23) is expected to be waked by the upstream turbine (11).

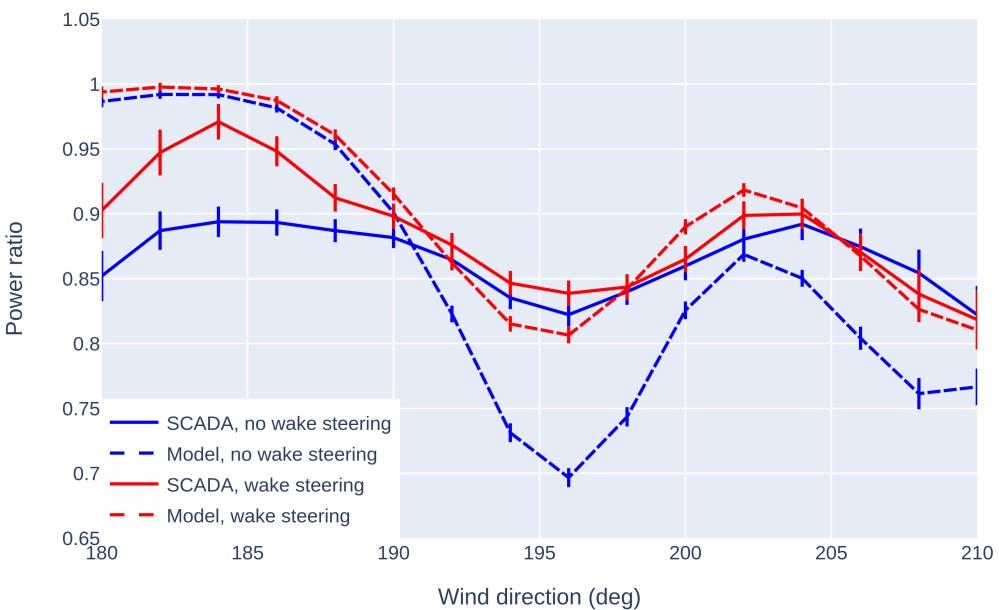

**Figure 11.** Power ratio versus wind direction for turbine 23 from the SCADA (solid lines) and the model (dashed lines). Results are shown both with wake steering control deactivated ("no wake steering") and with wake steering control activated ("wake steering"). Error bars correspond to 83% confidence intervals, such that marginally non-overlapping error bars approximate a 95% statistically significant difference between the means (Goldstein and Healy, 1995).

in a real world closed-loop application the model would be retrained more frequently, i.e., would not be used in the controller for months, or even weeks, without being retrained.

### 3.3 Aggregate metrics

We studied several metrics to compare the overall performance of the models: mean-squared error of the power signals, mean-squared error of the power ratio signals, correlation coefficient of the power signal, and correlation coefficient of the power ratio

signal. All metrics showed similar trends, so for ease of interpretation and comparison, we chose the correlation coefficient of the power ratio (in a 1 minute time series sense) as our main metric on which to judge model performance. The correlation coefficient of the power ratio is also strongly affected by modeling errors across the entire plant because incorrect power predictions at the reference turbines will affect the power ratio predicted at all turbines on the farm. The correlation coefficient

**Table 1.** A description of the parameters used for each model in this study. $C_P$ denotes the source of the power coefficient curve, with "OEM" indicating that the curves were determined from OEM specifications, and "SCADA" indicating that the curves were determined by fitting to historical SCADA data. "Wind direction source" gives the wind direction source. $\Delta\theta$ is the estimated wind direction offset. $\mathcal{N}_U$ and $N_\theta$ denote the wind speed and wind direction spatial filter widths, respectively. The "TI mapping" column indicates whether a TI mapping is used. (The details of the TI mapping for Model 0 are given in Sect. 2.4.5.) The "Output corrector" column indicates whether the output corrector is used.

| ID | $C_P$ | Wind direction source | $\Delta\theta$ | $\mathcal{N}_U$ | $N_\theta$ | TI mapping | Output corrector |
|----|-------|----------------------|----------------|-----------------|------------|------------|------------------|
| 0 | OEM | SCADA & GNSS compass | $0°$ | $\infty$ | 5 | Yes | No |
| 1 | SCADA | GNSS compass | $0°$ | $\infty$ | $\infty$ | No | No |
| 2 | SCADA | GNSS compass | $5°$ | $\infty$ | $\infty$ | No | No |
| 3 | SCADA | GNSS compass | $5°$ | 10 | $\infty$ | No | No |
| 4 | SCADA | GNSS compass | $5°$ | 10 | $\infty$ | No | Yes |

**Table 2.** The correlation coefficient of the power ratio for different models. Because different models will differ in the impact on freestream power estimation and waked power estimation, we decompose the dataset into different wind direction sectors and waking/waked turbine clusters. "Full wind sector, all turbines" shows overall performance. "Wind from the south, all turbines" focuses on all turbines for the waking wind sector of interest (wind directions from $152°$ to $242°$ and wind speeds from 5 to $12\ \mathrm{m\,s^{-1}}$). "Wind from the south, turbines 8–13" considers the performance of predicting upstream turbines that may wake the second-row turbines. "Wind from the south, turbines 19–24" examines the performance of predicting turbines that may be waked in the second row.

| Model Number<br>Data Subset | 0 | 1 | 2 | 3 | 4 |
|------------------------------|-------|-------|-------|-------|-------|
| Full wind sector, all turbines | 0.058 | 0.058 | 0.063 | 0.173 | 0.837 |
| Wind from the south, all turbines | 0.048 | 0.068 | 0.083 | 0.779 | 0.894 |
| Wind from the south, turbines 8–13 | 0.058 | 0.089 | 0.087 | 0.748 | 0.933 |
| Wind from the south, turbines 19–24 | 0.158 | 0.096 | 0.199 | 0.474 | 0.724 |

of the power ratio is computed for each model over different subsets of data and groups of turbines and shown in Table 2. Quantities were computed over the total validation time range (December 1, 2021 to April 1, 2022).

As expected, the initial naive model performance is generally poor, in agreement with our observations in Sect. 3.1. The remaining validations will therefore focus on Models 1–4. Starting from the baseline model (Model 1), the aggregate metrics generally improve with each addition to the modeling framework, and this improvement holds across most subsets of the data. The addition of the wind direction offset (Model 1 to Model 2) slightly improves the power ratio correlation coefficients for all turbines in the full wind sector. By examining the subsets of the data, we see that this change is driven by improvements to second-row turbines (19–24) and that front-row turbines (8–13) show negligible difference (the slight differences observed are likely just due to differences in the reference turbines, which change as the wind direction changes), as expected.

Allowing wind speed heterogeneity (Model 2 to Model 3) substantially improves the correlation coefficients for two reasons. The first is an improvement in modeling the reference turbine powers in the front row, which is evident from the large increase in correlation coefficients across upstream turbines 8–13, from $0.087$ to $0.748$. The second is an improvement in modeling the local ambient wind speeds that the engineering wake model uses to derive the wake deficit, which can be seen from the moderate increase in correlation coefficients across downstream turbines 19–24, from $0.199$ to $0.474$. Since Model 3 shows larger correlation coefficients on upstream turbines ($0.748$) than on downstream turbines ($0.474$), we conclude that this model is more successful in predicting upstream turbine power than downstream turbine power.

The output corrector model (Model 4) shows the best performance across all data subsets. The correlation coefficient for Model 4 is about $0.7$ to $0.9$ (depending on the dataset considered), indicating excellent agreement between this model and the observations, demonstrating the value of this hybrid approach to improve a model's understanding over time. Additionally, the greatly improved performance of Model 3 over Models 1 and 2 highlights the potential for deploying an initial model without an output corrector, providing reasonable accuracy while enough wake steering observations are collected to train the output corrector to the point where it can make predictions on the effects of various yaw misalignments.

### 3.4 Power ratio versus wind direction

Figure 12 shows the power ratio as a function of the SCADA wind direction when wake steering was disabled for two representative waked turbines: 19 and 24. The data has been filtered for region 2 wind conditions and for times when wake steering control was toggled off, yielding about 2 to 3 weeks of data over the 4 month measurement period. As in Fig. 11, we use overlapping bins ($4°$ bin width and $2°$ bin step) to reduce noise in the data.

Wind directions with low power ratios in Model 1 are offset from those in the SCADA, indicating that this model does not capture wake locations properly. (For example, Model 1 shows a trough in the power ratio at $188°$ in Fig. 12(a), whereas the SCADA shows a peak in the power ratio here.) By including the wind direction offset, Model 2 significantly improves predictions of wake locations, especially for turbine 19 in Fig. 12(a). However, we still observe misalignment in some peak and trough locations between Model 2 and the SCADA for turbine 24 in Fig. 12(b). (For example, Model 2 shows a peak at $188°$, while the SCADA shows a peak at $192°$.)

Model 3 (which introduces wind speed heterogeneity) shows similar peak and trough locations to Model 2, but with different power ratio magnitude values. This suggests that wind speed heterogeneity does not have a significant impact on downstream waking locations but may affect the expected power losses. While the overall metrics in Sect. 3.3 indicate that Model 3 gives better performance on downstream turbines, the power ratio results do not show a clear improvement in accuracy with Model 3. Model 4 shows the closest match in peak and trough locations in the SCADA, particularly in Fig. 12(b), indicating that the output corrector model gives the best prediction of waking and unwaking trends.

In Fig. 13, we consider these same plots for times when wake steering was enabled. This will allow us to assess if our model tuning process also improves the accuracy when upstream turbines may yaw to steer downstream wakes. The results mirror those with waking steering disabled. As before, we observe that the wind direction offset generally improves our modeling of wake locations (e.g., compare the power ratio predictions at $200°$ between SCADA, Model 1, and Model 2 in Fig. 13(a)), but

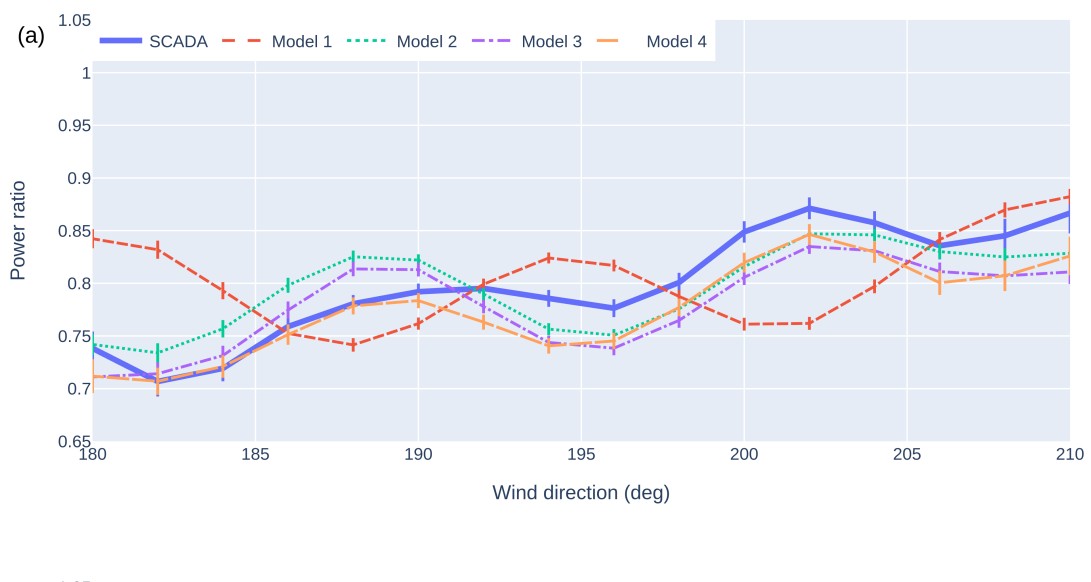

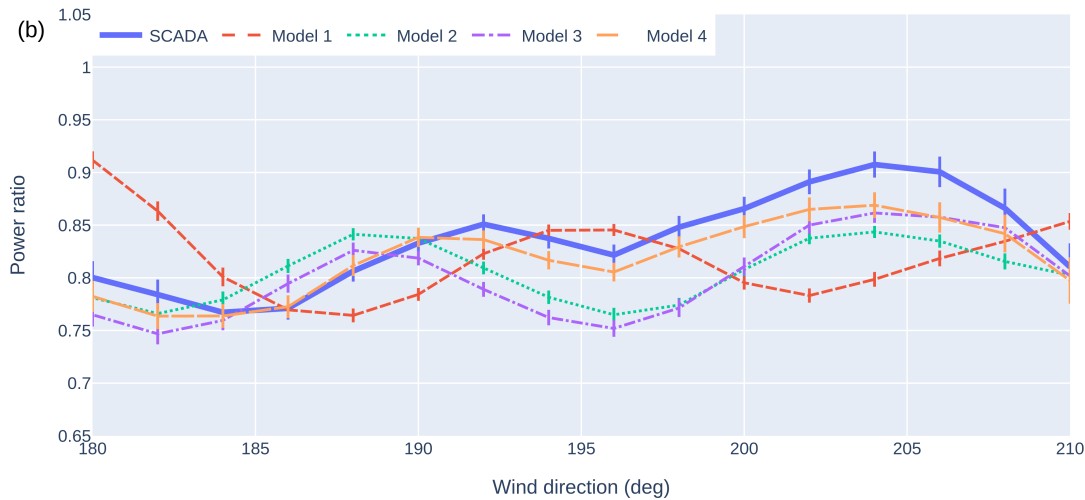

**Figure 12.** The power ratio as a function of turbine's SCADA wind direction from SCADA and various models for turbine 19 (a) and 24 (b) when wake steering is disabled. Error bars correspond to 83% confidence intervals, such that marginally non-overlapping error bars approximate a 95% statistically significant difference between the means (Goldstein and Healy, 1995). Models 2 and 3, which differ in wind speed heterogeneity, show similar patterns of waking behavior but different magnitudes. The output corrector model (Model 4) differs from other models in magnitude for both turbines and in phase for turbine 24. The baseline model (Model 1) shows a relatively large phase shift with respect to the SCADA data.

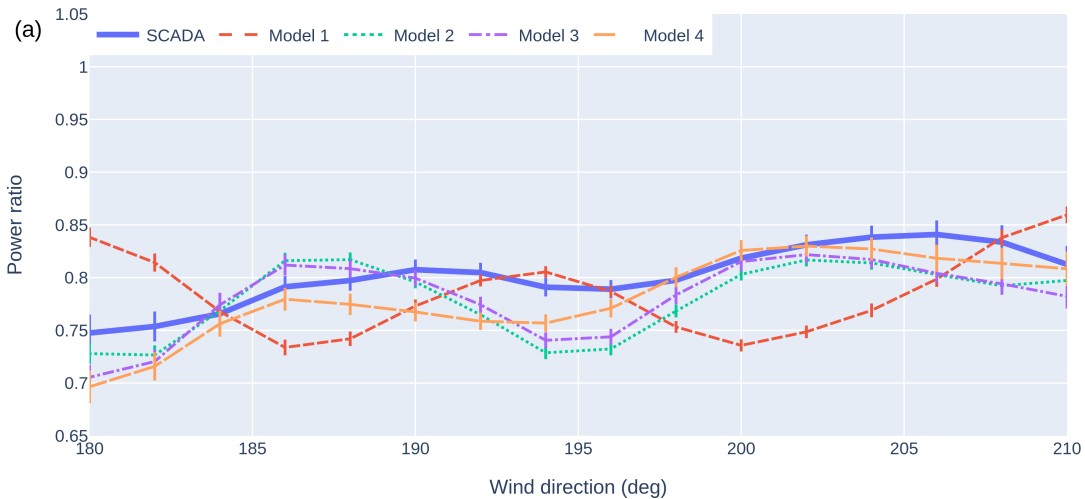

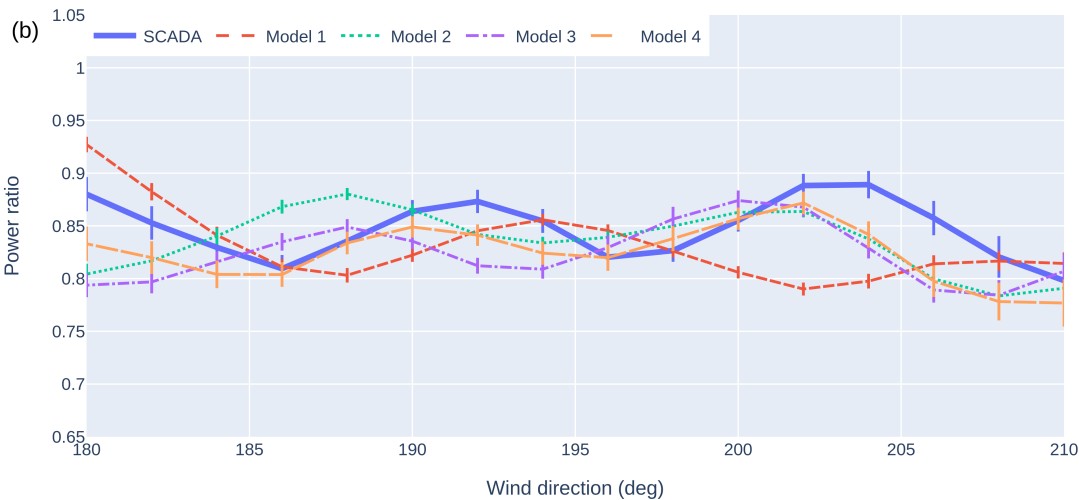

**Figure 13.** The power ratio as a function of turbine's SCADA wind direction from SCADA and various models for turbine 19 (a) and 24 (b) when wake steering is enabled. Error bars correspond to 83% confidence intervals, such that marginally non-overlapping error bars approximate a 95% statistically significant difference between the means (Goldstein and Healy, 1995).

there is some slight misalignment for turbine 24 near 190° in Fig. 13(b). Also as before, while the power ratio values differ between Models 2 and 3, the peak and trough locations are similar, and Model 3 does not represent a clear improvement from Model 2. The peak and trough locations from Model 4 most closely match the SCADA for both turbines 19 and 24, indicating that the output corrector model is expected to give the best predictions of wake locations, but discrepancies still exist when comparing Model 4 to SCADA values.

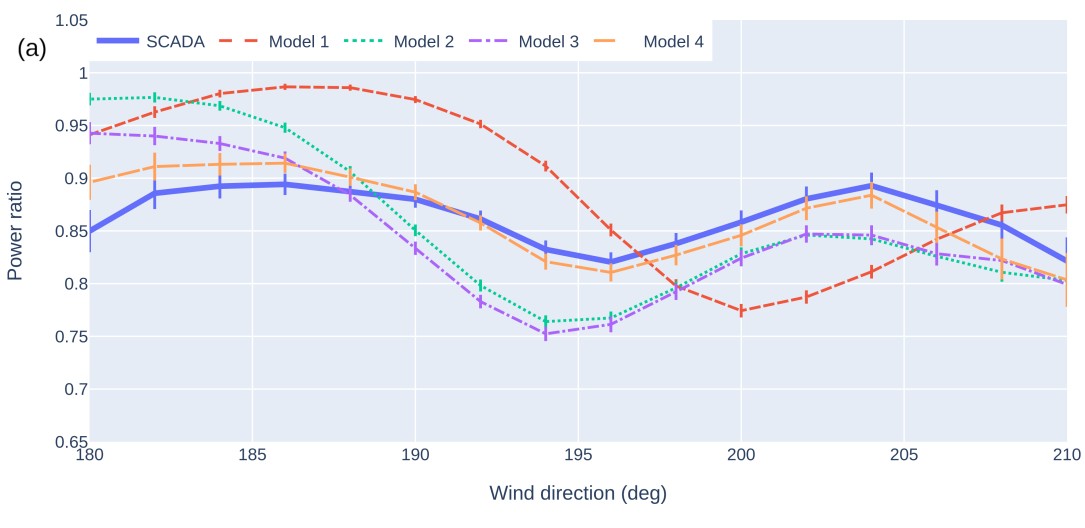

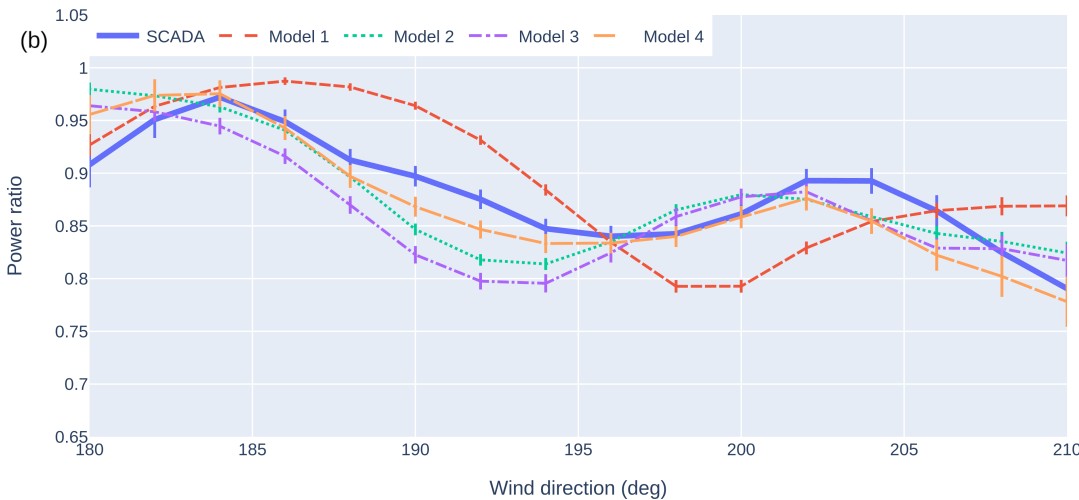

**Figure 14.** Turbine 23's power ratio as a function of its own SCADA wind direction computed with various models with wake steering disabled (a) and enabled (b), for comparison to the initial deployed model performance shown in Fig. 11.

Finally, Fig. 14 shows binned mean power ratio predictions versus wind direction for turbine 23, for comparison with the initial deployed model predictions shown in Fig. 11. We see again how Model 1 fails to predict the correct wake locations. Models 2 and 3 appear to overestimate the impact of waking and the benefit from wake steering, while Model 4 shows closest agreement with the observed performance.

## 3.5 Time series

We finally consider two time series in detail as an anecdotal examination of model performance. For the first time series, we consider a case without wake steering, and for the second time series, we consider a case with wake steering.

### 3.5.1 Time series without wake steering

Figures 15 and 16 compare SCADA results and model predictions for a representative time series when wake steering control was disabled. Figure 15 shows results for turbine 10 (an upstream turbine). As expected, adding the wind direction offset (Model 1 to Model 2) has negligible effect, since this turbine is unwaked. While the power predictions from Models 1–3 follow the same trend, Model 3 matches the SCADA power values most closely, indicating that introducing wind speed heterogeneity into the model improves the predictions at upstream turbines, in agreement with our findings in Sect. 3.3. Model 4 has the strongest correlation with the underlying SCADA time series and gives the most accurate power predictions, as expected, since it involves machine learning on real-world operational data. However, Model 4 predictions lag the measured power by about 1 minute and exhibit slightly higher power values throughout much of the time series. While the reason for the time lag is unclear, the biased power values indicate that this turbine produced higher power in the training period at comparable wind conditions. This bias is not expected to be an issue in a field deployment of this model architecture, since the deployed model would continually be retrained based on the latest available data and thus would be less affected by historical changes in turbine performance.

Model 4's discrepancies highlight the challenges of interpreting machine learning models, which would ideally have errors that are not correlated in time. In other words, a correctly specified model will have errors that appear to be independent and identically distributed, zero mean gaussian noise. When model errors (residuals) are correlated with one of the input features (time here), then there is typically room to improve the model.

While we can hypothesize about how the model is making predictions, it's more difficult to understand why the model converged to a particular solution. A strong case can be made that the time lagged target turbine's power is a strong predictor of the most recent power, and the model is just making small perturbations to the time lagged power signal (which is the intent of the feature space design). A case can also be made that the advection time is causing the observed time lag. However, the goal is to determine how to improve the model. But, it's unclear why the model is failing to converge to a non-time lagged solution in our current implementation, since it should have the necessary features to do so. Potential paths forward could be additional feature engineering, including additional time lags, collecting more training data, and/or adjusting the model hyperparameters.

Figure 16 shows results for turbine 22 (the turbine downstream of turbine 10). From Fig. 16(a), we see that this turbine will encounter one waking wind direction (horizontal dashed line) over this hour. Model 1 (which has no wind direction offset) predicts that we cross this wind direction at 23:45 and then transition to a wind direction with less waking (23:50 to the end). Models 2 and 3 (which include the $+5°$ wind direction offset) predict that we will reach this wind direction and remain there for the final 10 minutes of the time series.

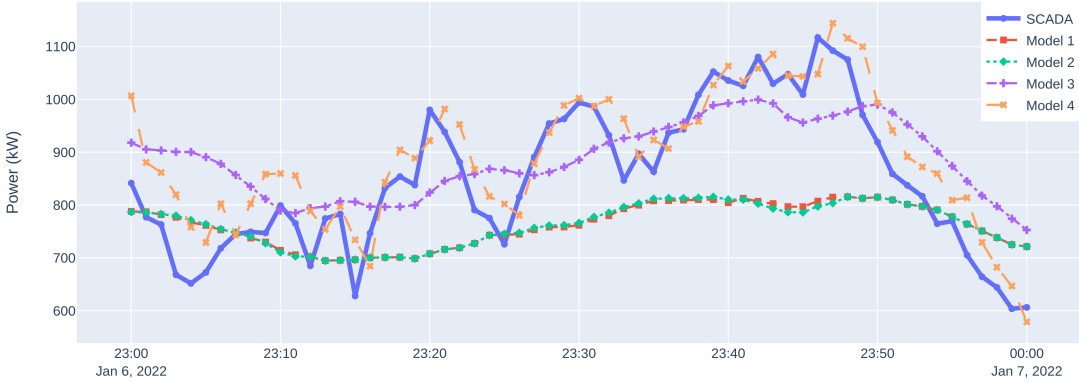

**Figure 15.** Example time series showing the power of turbine 10 when wake steering is disabled. Turbine 10 is a free stream turbine in this sector. SCADA results are compared against the predictions from various models from Table 1.

The power ratio and power predictions (Fig. 16(b) and 16(c), respectively) provide a good test of these models. Model 1 predicts that waking occurs from about 23:30 to 23:45 (power ratios below 1) and is not present near the end of the time series.
The SCADA data, however, show that waking occurs from about 23:45 to the end of the time series, highlighting the potential for detrimental control decisions if Model 1 were to be used in a wake steering controller. Model 2 captures this trend very well, suggesting that the wind direction offset has enabled us to successfully predict the dynamic onset of waking. Model 3 shows very similar results to Model 2 and thus that including wind speed heterogeneity has little effect on this downstream turbine (in contrast to the benefit it provided on the corresponding upstream turbine).
While Models 2 and 3 substantially improve waking predictions near the end of this time series, discrepancies remain between these models and the SCADA early in the time series. For example, at 23:15, the SCADA shows low power ratios and powers for this turbine, which do not seem to be explained by waking. Model 4, however, is generally able to capture these trends well, indicating that our output corrector approach can represent phenomena beyond what can be captured in an EWM (e.g., large levels of flow heterogeneity or rapid flow transients). As before, Model 4 has an apparent 1 minute time lag and
some biased power values, the latter of which may be due to changes in turbine performance from the training period. Again, this is a conservative test of the output corrector since it was not trained on the 2 month period that contains this time series.

### 3.5.2 Time series with wake steering

Figures 17 and 18 compare SCADA results and model predictions for a representative time series when wake steering was activated (the same time series considered in Fig. 10). Figure 17 shows results for turbine 11 (an upstream turbine that is wake
steering). As was observed in Sect. 3.5.1, adding the wind direction offset (Model 1 to Model 2) has negligible effect here, and introducing wind speed heterogeneity (Model 2 to Model 3) improves the power predictions. Model 4 again correlates well

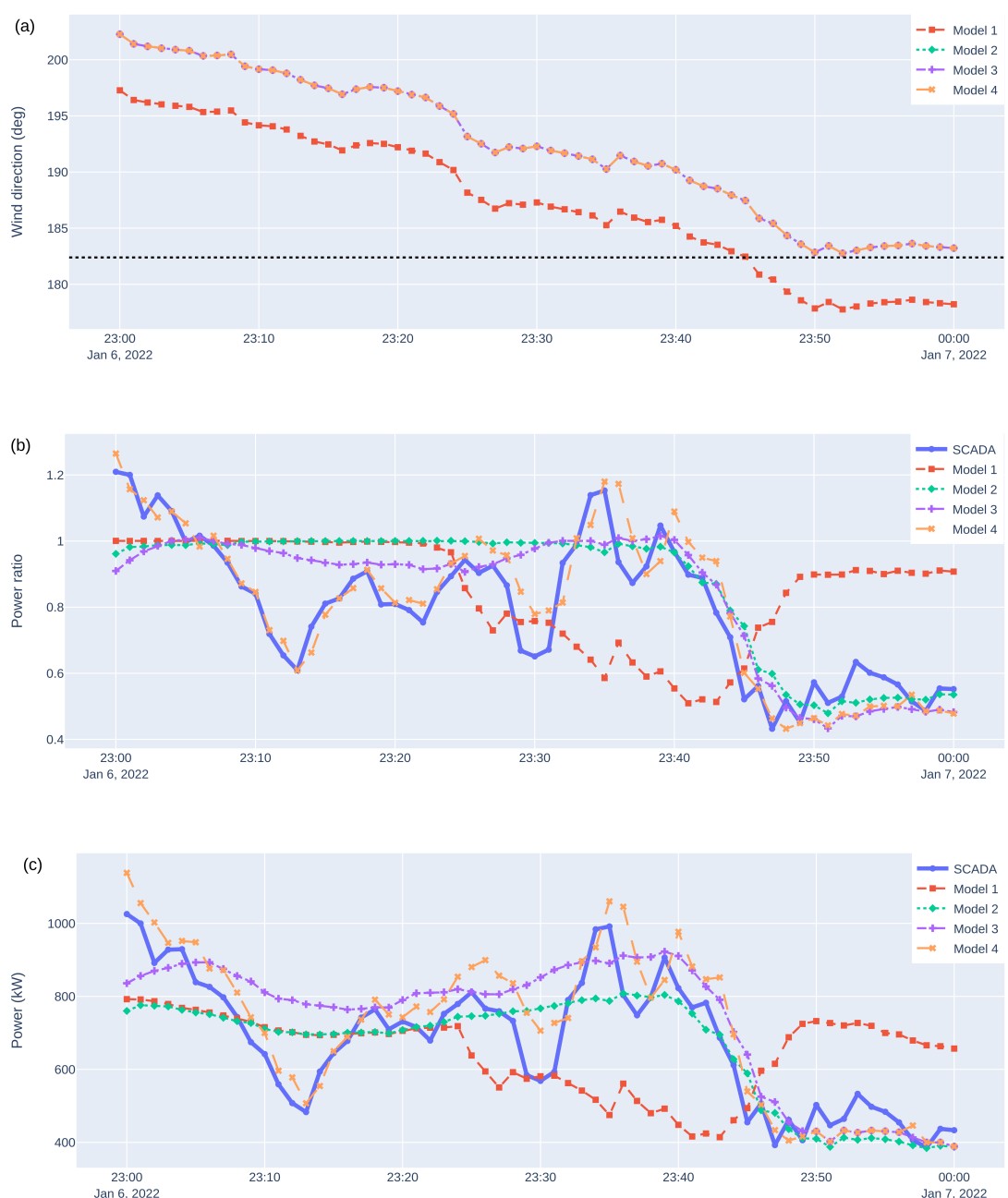

**Figure 16.** Example time series showing wind direction (a), power ratio (b), and power (c) for turbine 22 for a time when wake steering is disabled. SCADA results are compared against the predictions from various models from Table 1. The horizontal dotted line in (a) shows a wind direction where the downstream turbine (22) is expected to be waked by the upstream turbine (10). Several drops in power ratio occur throughout the time series, likely due to a combination of heterogeneity and waking on short time scales.

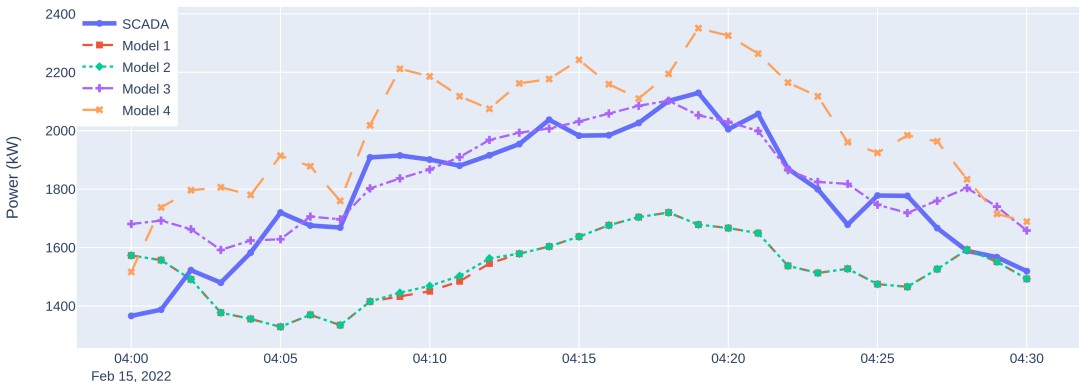

**Figure 17.** Example time series showing the power of turbine 11 when it is steering. SCADA results are compared against the predictions from various models from Table 1.

with the underlying SCADA signal; however, there is a slight bias to the power predictions that we attribute to different turbine performance in the training period, which could be addressed by retraining the model on the latest data.

Figure 18 shows results for turbine 23 (the turbine downstream of turbine 11). From Fig. 18(a), we see that Model 1 expects
turbine 23 to be significantly waked by turbine 11, and thus predicts a power ratio below 1 for the entire time range considered
(Fig. 18(b)). The SCADA data, however, shows that the power ratio fluctuates between a value near 1 (indicating no waking)
and a value below 1 (indicating waking). Models 2 and 3 include the $+5°$ wind direction offset, shifting the model wind
direction (Fig. 18(a)) above that at which we expect significant waking. As a result, the model power ratios are larger and
closer to the SCADA data over much of the time range. One exception is at the start of the time series (4:00), where the
SCADA shows much lower power values than the models. The cause of this discrepancy is unclear, since the average wind
directions and TIs (not shown) are similar to those at later times when there are higher powers (4:20). As before, we expect this
phenomenon to be driven by wind condition transients or heterogeneity levels that are not captured by our engineering wake
model. Finally, we see that allowing wind speed heterogeneity (Model 2 to Model 3), while changing the power and power
ratio predictions of the downstream turbine, does not substantially improve the model accuracy at the downstream turbine
for this particular time series. Model 4 shows strong correlation with the observations and successfully captures additional
information missed by Models 1–3. Though the dataset does not contain enough operation similar to this scenario to evaluate
the downstream turbine's energy gain in aggregate, this should be a focus of future work, along with the ability to predict the
upstream turbine's power loss, in order to the model's effectiveness at predicting the total effective of wake steering.

## 3.6    Suitability for optimization

Despite doing better at predicting the 1 minute behavior of the plant power, the models also must be assessed for their suitability
to be used with an optimization algorithm to find the optimum yaw values for all steerable turbines. Given that all models except

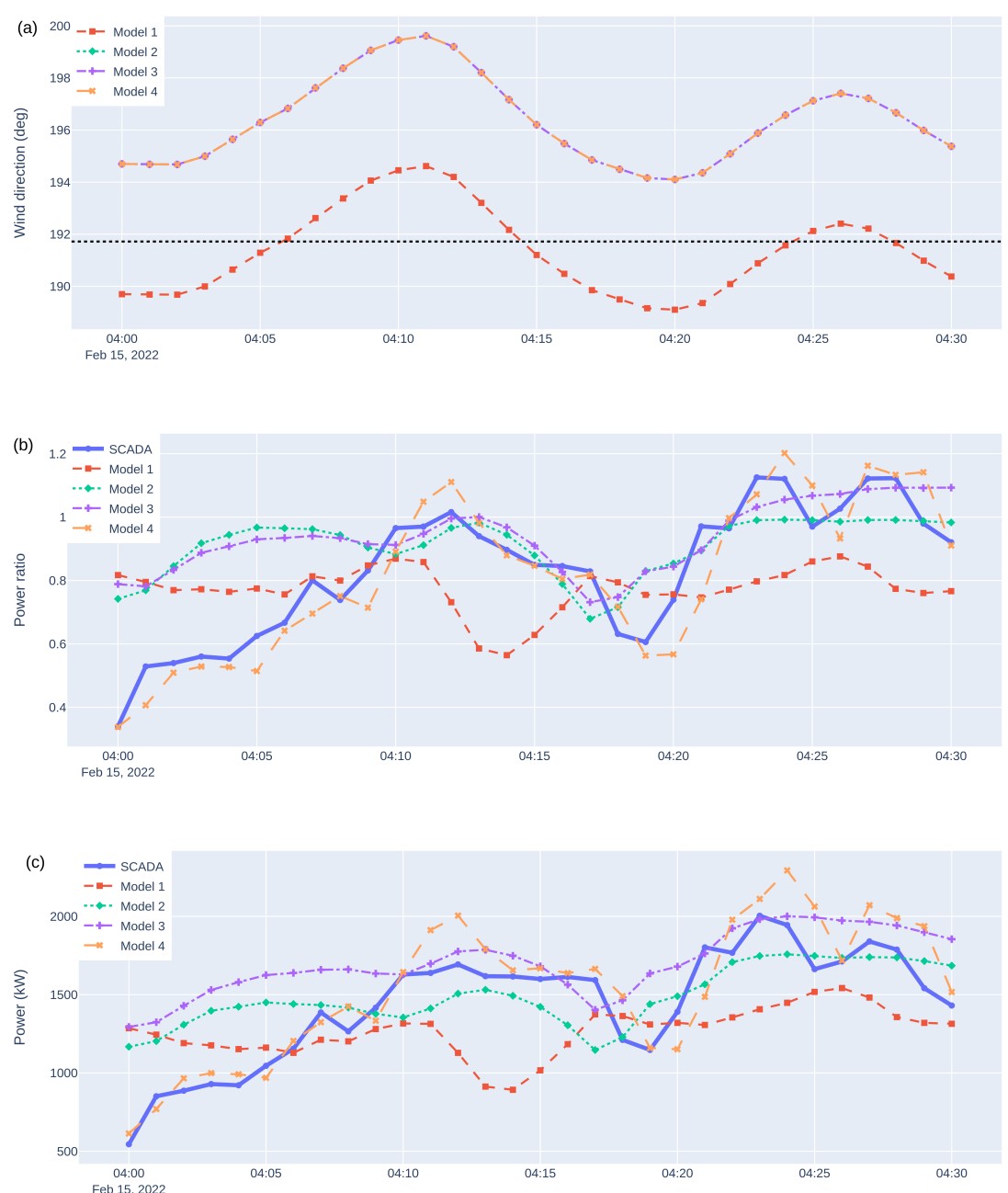

**Figure 18.** Example time series showing wind direction (a), power ratio (b), and power (c) for turbine 23, which is downstream of a turbine that is wake steering (turbine 11). SCADA results are compared against the predictions from various models from Table 1. The horizontal dotted line in (a) shows a wind direction where the downstream turbine (23) is expected to be waked by the upstream turbine (11).

Model 4 use an analytical formulation designed to be continuous for gradient-based optimization, it can be assumed that they will not cause any problems therein. However, Model 4 had to be tested to ensure its predictions of farm power remained smooth in the yaw angle dimension.

Figure 19 shows the resulting farm power gain predicted over a parameter sweep of turbine 13's yaw angle. We're looking at the predicted farm power gain due to yawing a single turbine. The numerator is the sum of each turbine's power caused by changing only turbine 13's yaw angle. The denominator is the sum of each turbine's power with all yaw angles set at zero. Behind the scenes, there is a suite of machine learning models being used to recompute the power at each individual turbine (one model per turbine). Only a small subset of downstream turbine's powers (hopefully one or two, depending on the wake width) depend on that free stream turbine's yaw error. Only one upstream turbine's power depends on that yaw angle (the turbine whose yaw angle is being adjusted). When that individual model reverts back to the engineering wake model, the predicted power for the affected subset changes significantly. The other turbine's power's remain at their adjusted values. So, between model 3 and 4, the numerator and the denominator are very different values, even when a single machine learning reverts back to using only the EWM. The difference in bias between the underlying EWM and the machine learning model is the cause of the discontinuity.

A wake steering controller would roughly want to find the maximum of this function in order to prescribe optimum yaw values. We see that there is a discontinuity in the results with the output corrector, highlighting the effectiveness of the feature space filter to prevent extrapolation beyond the training data. However, we also see the potential for "trapping" the controller output if the discontinuity were reversed, such that it might never prescribe larger yaw angles, and therefore the output corrector would not be able to learn from observations at these larger yaw angles. This is not the case in Figure 19, i.e., the feature-space-filtered EWM results would inspire a larger yaw angle, but this may not always be the case. To this end, further work may be required to ensure predictions are smooth in the yaw angle dimension, or to detect times when yaw angles are being limited by the discontinuity, such that the EWM can be used to further explore the feature space.

One final consideration is computational cost. The changes made to get to Model 3 from Models 1 and 2 do not appreciably increase computational cost, allowing a real time optimization to be computed in less than a minute. The output corrector, however, adds cost from model selection logic, neural network prediction, and feature space filtering. In the form tested here, this increases time to run a wake steering optimization several-fold. This is certainly not insurmountable given the potential for neural networks to use higher performance libraries, GPU hardware, and/or to store information about the gradient of the target with respect to features (namely yaw in our case), which negates the need to use a finite difference to compute gradients as in the implementation tested here.

## 4   Conclusions

A model-based wake steering control system was developed and deployed to 10 turbines on a 58 turbine wind farm with 13 rotor diameter spacing in the predominant wind direction—to the best of our knowledge the largest and most complex wake steering campaign to date. Wake steering was achieved through absolute nacelle position control, eliminating the shortcomings

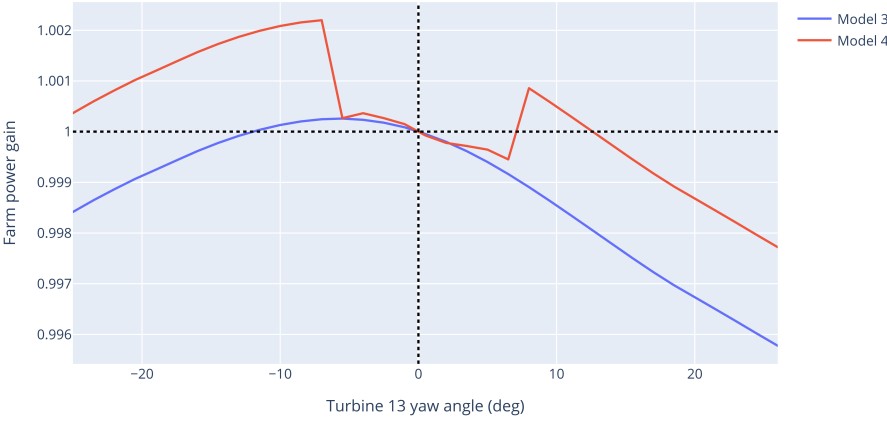

**Figure 19.** Predictions of farm power gain as a function of turbine 13's yaw angle with and without the output corrector (Model 4 and 3, respectively). For model 4, the full output corrector is used for yaw error values between −6 and +7 degrees. Outside of this range, all individual models in the ensemble will revert back to the engineering wake model, if that model depends on turbine 13's yaw error. All other models in the output corrector ensemble are used for the entire range.

of control via yaw offsets applied to a turbine's own measurement of the wind direction. Initial results from the pilot study showed power gains at the selected downstream turbines as a result of wake steering, though as expected, the preliminary minimally tuned model was not able to perfectly predict, and therefore optimize, the plant behavior. Based on the collected wake steering dataset, we developed a novel model architecture and validation approach to "close the loop" in order to improve the predictive capability, and therefore wake steering performance, over time. The validation approach included high-level

aggregate metrics (on the entire farm and on important subsets of the data), power ratio binned by wind direction for waked turbines, and detailed examination of key time series in the operational data.

     It was shown that the most simplistic model provided poor predictions of plant behavior, indicating the importance of this validation process for effective control. As complexity was added to the model, predictive capability improved. The most accurate model included a data-driven wind direction offset calibration, a Gaussian spatial filter to allow for wind speed

heterogeneity, and the neural network-based output corrector. However, even without the output corrector, it was shown that it is possible to achieve substantial improvements with data-driven engineering wake model input estimation/calibration.

     The output corrector results show great promise for how the rapidly advancing fields of artificial intelligence and machine learning can be leveraged to develop more effective wind farm control systems. In this study a relatively simple neural network architecture was employed, and some weaknesses of the approach were highlighted, namely the additional computational cost

to make predictions and the potential for discontinuous outputs between the neural networks and the engineering wake model in the space of the optimization actuation variable (yaw angle). However, there are a vast array of new avenues to explore in future work: introducing more complex neural network models designed for forecasting (such as long short-term memory

or transformers) to enable something of a "preview" of wind direction like that simulated in Sengers et al. (2023), imposing physical constraints on the models, or using learned gradient information for efficient optimization.

Future work should also focus on testing the model's ability to predict the change in downstream turbine power and power ratio due to wake steering—not only its absolute value—both in a time series sense and in aggregate. Once enough operational data has been accumulated, the overall energy gain of the entire control system should also be evaluated, taking into account upstream turbine losses due to yaw.

*Data availability.* Because this work was carried out as part of the development of a commercial product, data are not openly available, as
this would reduce the competitive advantage of WindESCo in the market.

*Author contributions.* PB, PI, and BB wrote the software to create and validate the wake models, along with the centralized wind farm control software that makes use of these wake models. PB, PI, and BB also wrote the code to process and visualize the data collected. CQ developed the feature space filtering and Gaussian consensus algorithms and code, and helped with machine learning model selection and training. DZ managed the installation and debugging of the hardware and software at the site, as well as the development of the turbine-level
software interfaces. MD secured the resources for and helped plan the project. All authors contributed to the writing and editing of the manuscript.

*Competing interests.* The authors declare no competing interests.

*Acknowledgements.* The authors would like to acknowledge Longroad Energy for their support in carrying out this project. The authors would also like to acknowledge Brendan Taylor and Nathan Post for their efforts in developing and installing the pilot system.

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
