# Peer review of "Development and validation of a hybrid data-driven model-based wake steering controller and its application at a utility-scale wind plant"

_Wind Energy Science, 2023_

## Referee Comment (RC2)

**Reviewer comments to „Development and validation of a hybrid data-driven model-based wake steering controller and its application at a utility-scale wind plant" by Peter Bachant et al.**

The study describes the design of a wake steering controller that is deployed at a subset of turbines in a commercial wind farm. The focus is firstly on the processing of input data for the applied flow model. Secondly on the improvement of the model performance and its validation. The study is well structured and describes the design step by step in a transparent way. This is scientifically useful, as there is no standardized methodology of how to use wake models (time scales, input data) for wind farm control purposes.

**Comments on the content:**

The authors rely on a novel approach, that does not need pre-generated LUTs for the control strategy. Has the performance of this approach be compared to a „conventional" LUT-based control? The question arises because field-tests and experiments have shown that also the conventional yaw schedules, albeit not capturing all possible input conditions combinations, are able to produce power gains. Is the performance of the novel method better?

Line 230: It would be good to clarify here that the GPS signal of the edge devices signal follows he yaw position of the turbines and not the wind vane signal (I am assuming), which is not exactly the same thing.

Section 2.5 While it is very useful that the authors are transparent with the design, the bullet lists in this section get a bit extensive. Maybe the bullet points would be better allocated in a table.

Section 2.4.5 Did the authors access the behaviour of TI for different times of the day or under different stabilities? E.g. binning for day/night regimes could be a way to understand the variability in this variable.

The authors chose a quite short time scale of 1 min for many environmental variables. It seems not entirely clear to me what role time delays play in this study. Are they accounted for in the model or model input? At 8 m/s the flow needs 160s to propagate 13 rotor diameters. Does e.g the power of upstream and downstream turbines show higher correlation after a time delay? Could this be also a reason of the discrepancy mentioned in line 550?

**Other Comments:**

Line 498 „to to"

---

## Author Response (AR1)

**Reviewer 1 response**

The authors thank the reviewer for their thorough and thoughtful comments and questions, which have inspired changes to the manuscript that significantly improve its quality. Please see below for responses to specific questions and requests.

> The main area where the paper can be improved is showing how well the wake models predict the change in power or energy from wake steering. The results focus on the ability of the different model variations to predict the power ratios of different sets of turbines relative to unwaked turbines, which is useful for general wake model validation. But since the goal of the models is to optimize yaw offsets for wake steering to increase power capture, the ability of the models to predict the change in power with wake steering compared to normal operation should be investigated in more detail. For example, Figs. 12 and 13 show the power ratios as a function of wind direction for two waked turbines with and without wake steering, but the expected change in power from wake steering is probably small for these turbines and it is hard to determine how accurately the models capture this change from the separate plots. Plots showing the measured and predicted difference between the power ratios for wake steering and normal operation cases would be more effective at showing how well the model captures the change in energy from wake steering. Further, similar power ratio plots for upstream turbines (where we'd expect a power loss from wake steering), combined upstream and downstream turbines, and larger clusters of turbines (e.g., turbines 9-12 and 19-24) could help validate the models in a wider variety of scenarios capturing both power losses and gains at different turbines from wake steering. Aggregate metrics such as the total power or energy gain over a range of wind directions could be helpful too.

This is a valid point. It is true that the ability of the model to predict wake steering changes is not tested thoroughly in this study, i.e., the ability to predict change is implied from the ability to predict both conditions in the absolute. We have one time series in figure 17 to show that the model can predict power and power ratio well in steering cases, but have not shown its effectiveness at predicting the *change* due to steering, nor have we shown the ability to predict the sum of upstream and downstream turbines to analyze the overall gain from wake steering. This particular dataset unfortunately does not have enough steering operation to answer that questions rigorously, but is the focus of ongoing work as the system has accumulated more time in operation. Text has been added to the manuscript to explain the need to test this in future work.

> 1. Pg. 1, ln. 18: Another reference to consider citing for the magnitude of predicted wake losses is Lee and Fields (2021): https://doi.org/10.5194/wes-6-311-2021, which shows typical losses between 5% and 20%.

This reference has been added to the paper.

> 2. Pg. 2, ln. 29: "non-waked inflow": Or more accurately, the downstream turbine will experience less wake overlap or a reduction in wake velocity deficits.

Thank you for this clarification. The manuscript has been updated accordingly.

> 3. Pg. 3, ln. 59: "the timescales on which wind characteristics change.": Can you elaborate on this? For example, the wind characteristics change too quickly for the extremum seeking controller to converge?

Essentially yes. There are more constraints that make extremum-seeking control challenging for the present case, and these have been added to the manuscript, along with the clarification regarding wind direction variability.

> 4. Pg. 3, ln. 70: "helps the system account for spatial variation in wind characteristics.": It is hard to tell what helps the system account for spatial variation in wind characteristics. Can you clarify what part of the discussion above you are referring to?

This has been clarified in the text. We were trying to describe the problem associated with control decisions being made with only local wind information. Since wind direction in particular can vary in space and time, it is not necessarily correct to assume the wind direction at an upstream turbine is equal to that of a downstream turbine, and this assumption could lead to suboptimal control, especially with the large row spacing studied here.

> 5. Pg. 5, ln. 100: "The farm layout presents a challenge for wake steering control": Another challenge worth mentioning here is that because of the long distance between turbines in the predominant wind direction, wake losses are expected to be relatively low (if that is the case), so there is probably not much room for wake steering to increase power.

As we see in figure 11, there is an approximate 17% wake loss for second row turbines, and there is wind direction dependence, which indicates there is room for improvement from wake steering. However, improvements will likely be limited to 5–10%, judging by the difference between the peaks and troughs in the power ratio. Comments along these lines have been added to the paper.

> 6. Section 2.4.1: When estimating the Cp and Ct curves using nacelle wind speed measurements, do you account for potential biases (that are also potentially turbine-specific) between the nacelle wind speed measurements and the true freestream wind speed (i.e., by determining and then applying a nacelle transfer function)? Can you discuss how these biases could affect your estimation of the Cp and Ct parameters?

If biases between the nacelle wind speed measurements and the true freestream wind speed are not turbine-specific, our fitting procedure for $C_P$ will automatically account for them. This is because we are using the nacelle anemometer readings themselves to determine both $C_P$ and to estimate the ambient wind speed. This was part of our motivation for deriving the $C_P$ curve from data rather than using that provided by the manufacturer.

If biases are turbine-specific, this approach could run into issues. However, as noted in the paper, fitting at an individual turbine level was generally impractical due to limited data for fitting at each turbine. Since all turbines were of the same model, it is less likely to pose a significant issue.

We have no independent way to verify the $C_T$ curves, and thus we are assuming that the wind speeds in the thrust coefficient tables match those from the nacelle anemometers.

We have added additional content to the paper to briefly discuss these points.

> 7. Eq. 3: The power loss is more accurately modeled in FLORIS by scaling the effective wind speed experienced by the turbine by cosine(gamma)^(p/3). This way power is inherently kept at rated power above the rated wind speed when the turbine yaws.

We thank the reviewer for the explanation, which we have adapted and added in the latest revision.

> 8. Pg. 9, ln. 187 "These can be excluded by filtering any turbine wind speeds...": Could you clarify whether you apply this filtering in your wind speed estimator?

We have added text to clarify that this filtering was not applied in the current study.

> 9. Pg. 10, ln. 230: "we also have access to the wind direction reported by GNSS": A GNSS measures the nacelle orientation, but how do you determine the wind direction from the GNSS?

We have updated the text to clarify that we use the GNSS nacelle orientation combined with the measured yaw error to compute the wind direction.

> 10. Pg. 11, ln. 237: "The wind direction obtained through this approach, however, may differ from that needed in the model...": If you are using GNSS wind direction measurements, which should already provide accurate nacelle orientation measurements relative to true north, why would further calibration be needed?

There are a few reasons why using GNSS wind direction measurements may not completely eliminate the need to further calibration: 1. There may be biases in the yaw misalignment values. For example, even if we have an accurate measurement of the nacelle direction, we may not have an accurate measurement of the yaw misalignment angles, causing our wind direction measurement to be incorrect. 2. There may be issues in the GNSS compass orientations; e.g., the compasses may be installed a few degrees offset from the nacelle direction. 3. Features (e.g., mountains) or physical processes (e.g., Coriolis effects) not present in the model may cause the wakes to deflect as they move downstream. One way to handle this is to modify the wind direction in the model so that the wakes arrive at the desired location downstream in the model.

> Also, is the "manual" calibration of SCADA signals mentioned in the previous paragraph the same process as described in the rest of this section?

We've added text to clarify how this "manual" calibration was completed.

> 11. Pg. 11, ln. 239 and pg. 12, ln. 271: "waking directions":
>     How is "waking direction" different from "wind direction"
>     as used here? If there is no difference, consider using "wind
>     direction" to avoid confusion.

The meaning of "waking direction" is a bit different between the two examples noted and is indeed unclear. We have updated the text so that in the first instance, we have "the wind direction to best capture waking", and in the second instance, we have used "waking wind directions."

> 12. Equation 9: What temporal averaging period do you use
>     when calculating the power ratios?

We use five-minute temporal averaging period and have now added text to clarify this.

> 13. Pg. 12, ln. 255: "we filter for yaw errors below 5 degrees":
>     Do you require all of the reference turbines to have yaw
>     errors less than 5 degrees as well, or only the target turbine?

We require both the target and reference turbines to meet all the specified filtering criteria (including having yaw errors below 5 degrees). We have added text to clarify this.

> 14. Section 2.5: To confirm, how exactly is the combined
>     EWM/output corrector model intended to be used for opti-
>     mization? Are the input features measured at the current
>     1-minute time step k used to find the optimal yaw offsets
>     to be implemented during 1-minute period k+1?

That is the intention. Our current set up uses a 30 minute data buffer to estimate the features from the model at time $k$, at which point our optimization should take less than a minute to run. The optimal set point is determined roughly 60 seconds after the data is collected, and held constant over the next minute of operation.

> 15. Pg. 16, ln. 345: "we have mTk possible features": What
>     does "T" represent?

"T" represents the number of time lags considered in the feature space. This variable definition was not included in the original manuscript, and it has been

added.

16. Pg. 17: Why aren't any EWM inputs used as features in the models? Were they not helpful in improving estimation accuracy?

This is an excellent question, and certainly a topic that can be explored much more deeply than was done in this paper.Much of the feature space exploration was limited by the computational expense of training and testing such a large number of models on large datasets. There are a very large number (infinite?) of potential features we could use here, and we limited the selection based on the following logic:

The EWM inputs are smoothed to allow EWM to have an acceptable input. Too much spatial heterogeneity in the wind speed or wind direction inputs appear to cause EWMs flow fields to deteriorate due to numerical issues. If the heterogeneity is physically meaningful, but not representable with an EWM, then the goal of the output corrector is to learn this behavior and correct for the over simplified boundary conditions used in the EWM. We do, however, include the EWM's power prediction at each turbine, which is strongly correlated with the EWM's inputs.

17. Pg. 17, ln. 389: "1 minute rolling average of SCADA power" is listed twice.

Thank you for catching this. We've amended the list to be consistent with the feature set listed in our code.

18. Pg. 17, ln. 392: "but with the target turbine's wind speed and power measurements omitted". Are the 1-minute lagged power measurements at the target turbine omitted as well, or are they still used as features?

The 1-minute lagged power measurements at the target turbine are still used as features. The goal here is to capture any rotor inertia effects that are missed in the steady state approach. However, as you can guess, there is some risk that the model simply uses the previous power measurement as the next prediction! This does become an important model features, but we do see evidence that the model is not simply time shifting the signal.

19. Pg. 18, ln. 405: "was reduced by a factor of 8 from the IEC value": Why did you choose to make the waked sectors so narrow? This seems too narrow to capture the true width

of the wakes during higher turbulence intensities.

This is a challenging parameter to select. If it is much higher, then the waked sectors start to overlap too strongly and they are merged together into larger partitions. At a certain point, the partitioning deteriorates to a single bin. Using small bins results in a more targeted model and less features available for training. But, it increases our dataset size, because if we have a large number of features, then we need to discard any rows that have missing values. The more features, the more likely we are to have a missing value! So, decreasing the bin width helps in this regard.

Ultimately, the deciding factor needs to be the suitability of the model for control. We found that models trained with a large number of correlated features tended to struggle to find which upstream turbine was responsible for the waking. This was evident when we adjusted the yaw error signal of a turbine that should (in a given wind sector) have no effect on a particular down stream turbine, but the model predicted a power change anyway. By switching to smaller sectors, we were able to eliminate this effect, as far as we could tell with the testing performed.

A future concept could be to allow for over lapping bins (no longer partitions), and then use a weighted ensemble of model predictions to determine what the most likely candidate is. As for the effect at higher turbulence intensities, we do not anticipate much benefit for tuning a model for accuracy at higher TIs.

> 20. Pg. 18, ln. 416: "Any column that was missing more than 60% of the data...": Can you clarify what columns and rows correspond to in the feature matrix? Does this mean that features are removed if they are missing for more than 60% of the turbines, and then turbines are removed if any of their features are missing?

Thanks for catching this. We often use columns/features interchangeably due to our data format, and that is not clear. Columns in this context refer to features, and rows refer to individual timestamps of data. We've updated the language to reflect this in the paper. Features are removed if they are missing for more than 60% of the timestamps, which will happen for every model that considers that feature in its training.

That filtering is done on a per-model basis, but since each model uses the same dataset, it will apply to any turbine which considers using that feature.

> 21. Pg. 18, ln. 416: Additionally, can you discuss how the output predictor model deals with missing input features

during the training and prediction stages? As stated earlier, a maximum of 404 features can be used to predict waked turbines' power. It seems likely that some of this data would be missing a significant amount of the time. Can the model still be used to predict power if it uses a different combination of available input features than were used for training? And can the training stage be performed using different combinations of available input features for each 1-minute sample?

We've added additional text to clarify this point, and discussed how our dimensionality reduction algorithm helps address the issues caused by a high frequency of missing data occurring. In short, we do no imputation, and all features used during training must be available, otherwise we revert that model back to the EWM. Our dimensionality reduction algorithm creates a large number of models which depend on relatively few features, which reduces the likelihood that any specific model will encounter a missing feature. This is in contrast to a model like LightGBM, which automatically imputes missing data during prediction, which we deemed too risky for control applications of physical systems.

22. Pg. 18, ln. 430: "the model should accurately capture the mean power loss due to waking as a function of wind direction, wind speed, and TI.": In addition to wind direction, wind speed, and TI, the model should be able to predict power as a function of yaw misalignment. How is this validated?

Standard techniques in machine learning can be used to validate the model predictions in an aggregate sense and on individual predictions (e.g., residual analysis). In addition we can take any given data point and synthetically modify the yaw error of a particular turbine to study the effect of that feature on the model predictions. However, performance cannot be guaranteed outside of the region covered by the training dataset. Figure 18 in the original manuscript shows how the model predictions change with yaw misalignment, and it shows the discontinuity caused by the reversion back to FLORIS (at a turbine where no high yaw error data is available). Handling this discontinuity can be achieved through additional data collection (e.g., a design of experiments potentially).

23. Pg. 22, ln. 489: "The correlation coefficient of the power ratio is also strongly affected by modeling errors across the entire plant because incorrect power predictions at the reference turbines will affect the power ratio predicted at all turbines": Wouldn't some errors in power prediction

> tend to cancel out across the farm when using the power
> ratio metric? For example if power is underpredicted at all
> turbines in the farm (both unwaked and waked), then the
> errors would tend to cancel out when calculating the power
> ratio.

In our experience, it depends on the particular type of error. Some simple model biases are likely to cancel out (e.g., all power curves are biased the same for each turbine and the bias is constant across wind speeds). But, a counter example is when a turbine's power curve is only biased in the knee of the power curve. If one turbine is firmly in region two and another is in the knee of the power curve, then one turbine's prediction is unbiased and the other turbine's prediction is biased, and they will not cancel out.

When operating on large scale plants, we've seen evidence that significant noise is introduced into the power ratio signal due to advection times, large scale spatial heterogeneity, measurement errors, and model biases.

> 24. Table 2: Why does the correlation coefficient for turbines
>     8-13 decrease between models 1 and 2? The only change
>     in the model appears to be adding a 5-degree offset to the
>     wind direction, and these turbines are unwaked anyway.

While these turbines are unwaked, it's likely that adding a 5-degree offset changes the reference turbines used when computing the power ratios, which is responsible for the slight change in the power ratio correlation coefficient metrics.

> 25. Pg. 27, ln. 550: "While the reason for the time lag is
>     unclear..." and pg. 29, ln. 571: "Model 4 has an apparent
>     1 minute time lag...": The model features include the
>     target turbine's 1-minute rolling average of SCADA power
>     lagged by one minute for freestream turbines (and maybe
>     for waked turbines? (see comment 18)). It seems likely that
>     this lagged power feature is one of the best predictors of
>     power during the next 1-minute period, so the model heavily
>     relies on this measurement, explaining the time lag in the
>     prediction time series. Can you discuss this possibility in
>     the paper? Additionally, for waked turbines, the current
>     1-minute averaged power of freestream turbines are used as
>     features. Since the wind speeds at the freestream turbines
>     could tend to travel downstream at the mean wind speed
>     and interact with the waked turbines after some time delay,
>     could this behavior also contribute to the visible prediction

> time lag?

These are all excellent points, and we've added a more detailed discussion on this point to the paper. In short, hypothesizing about how a model is making predictions (e.g., importance of the time lagged power) is different than hypothesizing about why the model converged to that result in the first place. The latter is what we're attempting to highlight here: we gave the model multiple time lagged features to allow the model to combine that information to obtain a non-lagged prediction. But, the model sometimes converges to a lagged solution anyway, and it's unclear why.

> 26. Fig. 18: Please discuss this plot further. Is model 4 only trained with yaw angle magnitudes > ~8 degrees, so the model reverts to the EWM model for smaller yaw angles? Why is the predicted power gain so much higher for model 4 than model 3? The gain above 1 appears to be an order of magnitude higher with model 4 than model 3.

There is a lot going on behind the scenes in this plot–thank you for pointing out the lack of clarity. We've updated the discussion to add additional details.

**Reviewer 2 response**

The authors thank the reviewer for their thorough and thoughtful comments and questions, which have inspired changes to the manuscript that significantly improve its quality. Please see below for responses to specific questions and requests.

> The authors rely on a novel approach, that does not need pre-generated LUTs for the control strategy. Has the performance of this approach be compared to a „conventional" LUT-based control? The question arises because field-tests and experiments have shown that also the conventional yaw schedules, albeit not capturing all possible input conditions combinations, are able to produce power gains. Is the performance of the novel method better?

This is a great question. The EWM could be replaced with a LUT, but to achieve the level of versatility in this model, it would need to have many dimensions: - Wind speed (one dimension per turbine if heterogeneous) - Wind direction (one dimension per turbine if heterogeneous) - TI - One dimension for every turbine's power limit - One dimension every turbine's online state Further, most LUTs

are implemented at the turbine level, so the dimensions that involve more global information could not exist.

Implementing heterogeneous wind speed and direction would produce even more dimensions. As we've seen, these greatly impact model accuracy.

If we assume 10 steps in wind speed, 360 in wind direction, 10 in TI, and 10 turbines, a heterogeneous wind input LUT would have 360 million rows.

For a full farm implementation, where multiple rows of turbines can potentially interact with each other, the LUT would grow to 400 billion rows.

Regarding which approach will produce the best optimization, keeping input estimation the same, a many-dimensional LUT should do the same as an EWM. Further, the output corrector could be employed to improve the lookup table to improve optimization performance in a similar way. However, this would increase deployment time, since every time the output corrector is retrained, the entire LUT must be repopulated.

Comparing the achieved energy gain between a simple LUT and more complex model-based controller has been studied in Howland et al. (2022), and this has been noted in the manuscript.

> Line 230: It would be good to clarify here that the GPS signal of the edge devices signal follows he yaw position of the turbines and not the wind vane signal (I am assuming), which is not exactly the same thing.

The text has been updated to clarify this point.

> Section 2.5 While it is very useful that the authors are transparent with the design, the bullet lists in this section get a bit extensive. Maybe the bullet points would be better allocated in a table.

The features have been condensed into a comma-separated list for compactness.

> Section 2.4.5 Did the authors access the behaviour of TI for different times of the day or under different stabilities? E.g. binning for day/night regimes could be a way to understand the variability in this variable.

We examined the turbulence intensity and found that it tended to be lower at night and higher during the day, though there was variability between different days and nights. We have not included that discussion here, however, since this focuses on fitting the turbulence intensity in the model rather than observations of the turbulence intensity in the field.

> The authors chose a quite short time scale of 1 min for many environmental variables. It seems not entirely clear to me what role time delays play in this study. Are they accounted for in the model or model input? At 8 m/s the flow needs 160s to propagate 13 rotor diameters. Does e.g the power of upstream and downstream turbines show higher correlation after a time delay? Could this be also a reason of the discrepancy mentioned in line 550?

Time delays have not been studied in great detail here. We are making a quasi-steady assumption, seeking a model that can accurately describe the power production of the plant at the current time, and therefore optimize yaw angles for that present time. Using longer lags or lagging other features could provide better accuracy, but we feel that this is good grounds for future work.

It is likely that the ideal controller can optimize forward into the future over a finite time horizon, taking into account yaw motion and convective delays, but we also chose to defer this complexity for future work. Since it takes on the order of a minute for a turbine to yaw to actuate wake steering, this at least provides the lower limit for how fast a controller should update.

> Line 498 „to to"

We have corrected this typo and thank the reviewer for bringing it to our attention.

---

## Author Response (AR2)

**Reviewer comment 2, reviewer 1**

1. Original reviewer comment: The main area where the paper can be improved is showing how well the wake models predict the change in power or energy from wake steering. The results focus on the ability of the different model variations to predict the power ratios of different sets of turbines relative to unwaked turbines, which is useful for general wake model validation. But since the goal of the models is to optimize yaw offsets for wake steering to increase power capture, the ability of the models to predict the change in power with wake steering compared to normal operation should be investigated in more detail. For example, Figs. 12 and 13 show the power ratios as a function of wind direction for two waked turbines with and without wake steering, but the expected change in power from wake steering is probably small for these turbines and it is hard to determine how accurately the models capture this change from the separate plots. Plots showing the measured and predicted difference between the power ratios for wake steering and normal operation cases would be more effective at showing how well the model captures the change in energy from wake steering. Further, similar power ratio plots for upstream turbines (where we'd expect a power loss from wake steering), combined upstream and downstream turbines, and larger clusters of turbines (e.g., turbines 9-12 and 19-24) could help validate the models in a wider variety of scenarios capturing both power losses and gains at different turbines from wake steering. Aggregate metrics such as the total power or energy gain over a range of wind directions could be helpful too.

Author response: This is a valid point. It is true that the ability of the model to predict wake steering changes is not tested thoroughly in this study, i.e., the ability to predict change is implied from the ability to predict both conditions in the absolute. We have one time series in figure 17 to show that the model can predict power and power ratio well in steering cases, but have not shown its effectiveness at predicting the change due to steering, nor have we shown the ability to predict the sum of upstream and downstream turbines to analyze the overall gain from wake steering. This particular dataset unfortunately does not have enough steering operation to answer that questions rigorously,

but is the focus of ongoing work as the system has accumulated more time in operation. Text has been added to the manuscript to explain the need to test this in future work.

It may be true that the particular dataset doesn't include enough wake steering operation to rigorously determine how well the models capture the change in power from wake steering. But some of the existing figures suggest that examples could be shown. In Fig. 11, the power ratios of a waked turbine with and without wake steering are compared to predictions from the initial model (revealing deficiencies in the initial model). A similar plot, comparing the SCADA power ratios with and without wake steering to the best-performing, more sophisticated model(s) would be insightful and illustrate how much better the more advanced models are at predicting the change in power from wake steering. This could be plotted using the data already contained in Figs. 12 and 13. The power ratios in Figs. 12a and 13a could be combined in a single plot to show how well models 1-4 predict the change in power from wake steering (though only the best-performing model(s) could be shown to reduce clutter in the plots).

We have added figure 14 to show these results from all models, for both wake steering enabled and disabled. We believe that this helps illustrate the improvements compared to figure 11.

2. Original reviewer comment: Pg. 1, ln. 18: Another reference to consider citing for the magnitude of predicted wake losses is Lee and Fields (2021): https://doi.org/10.5194/wes-6-311-2021, which shows typical losses between 5% and 20%.

Author response: This reference has been added to the paper.

Reviewer comment 2: Although the introduction says that onshore wake losses for US wind plants have been estimated to be between 2 and 20%, it is worth noting that the values up to 20% in Lee and Fields (2021) are not necessarily for onshore plants.

We have clarified the wording to ensure it is clear that 20% loss is not necessarily for onshore plants, but can include offshore as well.

3. Original reviewer comment: Section 2.4.1: When estimating the Cp and Ct curves using nacelle wind speed measurements, do you account for potential biases (that are also potentially turbine-specific) between the nacelle wind speed measurements and the true freestream wind speed (i.e., by determining and then applying a nacelle transfer function)? Can you discuss how these biases could affect your estimation of the Cp and Ct parameters?

Author response: If biases between the nacelle wind speed measurements and the true freestream wind speed are not turbine-specific, our fitting procedure for CP will automatically account for them. This is because we are using the nacelle anemometer readings themselves to determine both CP and to estimate the ambient wind speed. This was part of our motivation for deriving the CP curve from data rather than using that provided by the manufacturer.

If biases are turbine-specific, this approach could run into issues. However, as noted in the paper, fitting at an individual turbine level was generally impractical due to limited data for fitting at each turbine. Since all turbines were of the same model, it is less likely to pose a significant issue.

We have no independent way to verify the CT curves, and thus we are assuming that the wind speeds in the thrust coefficient tables match those from the nacelle anemometers.

We have added additional content to the paper to briefly discuss these points.

Reviewer comment 2: I agree that any Cp estimation errors caused by biased nacelle wind speed measurements would "cancel out" when calculating turbine power using the estimated Cp and nacelle wind speed measurements. However, it is worth noting that the Cp estimates themselves may be unrealistic. For example, if the nacelle wind speed measurement is much lower than the true freestream wind speed, the estimated Cp could be

> unrealistically high ($> 0.59$). Still, as you mention, this would not affect the accuracy of the final absolute power estimate.

> Additionally, in Fig. 5, are the legend labels switched? The curve labeled "from manufacturer" is "noisier" than the other curve, but I would expect the historical data-derived curve to be noisier.

A comment has been added to the manuscript regarding the possibility of unrealistic Cp values.

The legend labels are indeed correct, though there is a minor typo in our transcribed power curve at 6.5 m/s. This has been left as-is and noted in the figure caption since this was the power curve input to the initial deployed model.

> 4. Original reviewer comment: Pg. 17, ln. 392: "but with the target turbine's wind speed and power measurements omitted". Are the 1-minute lagged power measurements at the target turbine omitted as well, or are they still used as features?

> Author response: The 1-minute lagged power measurements at the target turbine are still used as features. The goal here is to capture any rotor inertia effects that are missed in the steady state approach. However, as you can guess, there is some risk that the model simply uses the previous power measurement as the next prediction! This does become an important model features, but we do see evidence that the model is not simply time shifting the signal.

> Reviewer comment 2: Thanks for the explanation. I would simply suggest clarifying that the 1-minute lagged power measurements at the target turbine are still used (i.e., only the current measurements are removed), for example by adding "but with the target turbine's"current 1-minute rolling average" wind speed and power measurements removed" (or similar).

The manuscript has been updated with this clarification.

5. Original reviewer comment: Fig. 18: Please discuss this plot further. Is model 4 only trained with yaw angle magnitudes > ~8 degrees, so the model reverts to the EWM model for smaller yaw angles? Why is the predicted power gain so much higher for model 4 than model 3? The gain above 1 appears to be an order of magnitude higher with model 4 than model 3.

Author response: There is a lot going on behind the scenes in this plot–thank you for pointing out the lack of clarity. We've updated the discussion to add additional details.

Reviewer comment 2: The added discussion indeed helps clarify the plot, especially how for yaw angles when the output corrector is disabled in Model 4, other turbines which are not affected by turbine 13 still use the output corrector. I had originally thought that the output corrector would have been disabled for all turbines in the farm.

My understanding now is that in the yaw angle sector from ~-6 to +7 degrees, the output corrector is used and outside of this sector Model 4 reverts to the EWM-only predictions for turbines influenced by turbine 13. Since I found this confusing initially I would suggest mentioning in the figure caption which yaw angles correspond to Model 4 using the output corrector and which correspond to the model reverting to the EWM predictions.

This suggestion has been added to the figure caption.